# The SUN-family protein Sad1 mediates heterochromatin spatial organization through interaction with histone H2A-H2B

Wenqi Sun[1,2,6], Qianhua Dong[3,6], Xueqing Li[1,2,6], Jinxin Gao[3,6], Xianwen Ye[2,4], Chunyi Hu [5], Fei Li [3] ✉ & Yong Chen [1,2,4] ✉

Heterochromatin is generally associated with the nuclear periphery, but how the spatial organization of heterochromatin is regulated to ensure epigenetic silencing remains unclear. Here we found that Sad1, an inner nuclear membrane SUN-family protein in fission yeast, interacts with histone H2A-H2B but not H3-H4. We solved the crystal structure of the histone binding motif (HBM) of Sad1 in complex with H2A-H2B, revealing the intimate contacts between Sad1(HBM) and H2A-H2B. Structure-based mutagenesis studies revealed that the H2A-H2B-binding activity of Sad1 is required for the dynamic distribution of Sad1 throughout the nuclear envelope (NE). The Sad1-H2A-H2B complex mediates tethering telomeres and the mating-type locus to the NE. This complex is also important for heterochromatin silencing. Mechanistically, H2A-H2B enhances the interaction between Sad1 and HDACs, including Clr3 and Sir2, to maintain epigenetic identity of heterochromatin. Interestingly, our results suggest that Sad1 exhibits the histone-enhanced liquid-liquid phase separation property, which helps recruit heterochromatin factors to the NE. Our results uncover an unexpected role of SUN-family proteins in heterochromatin regulation and suggest a nucleosome-independent role of H2A-H2B in regulating Sad1's functionality.

Eukaryotic heterochromatin is the transcriptionally silenced chromatin region and plays important roles in gene expression regulation and genome stability[1,2]. Heterochromatin is generally composed of repetitive DNA and is characterized by a unique epigenetic environment, including the methylation of histone H3 at lysine 9 (H3K9) and hypoacetylation of histones[1,2]. Heterochromatin regions are often associated with the nuclear periphery, a phenomenon conserved from yeasts to mammals[3,4]. However, how the spatial organization of heterochromatin is regulated and the biological importance of this organization remain poorly understood.

Fission yeast (*Schizosaccharomyces pombe*) has proven to be a powerful system for studying heterochromatin organization. Heterochromatin in *S. pombe* preferentially assembles at peri-centromeres, telomeres, and the mating-type (*mat*) locus[5,6]. H3K9 methylation in fission yeast heterochromatin is mediated by the Clr4 methyltransferase complex (CLRC), which plays a key role in the nucleation, spreading, and maintenance of heterochromatin[7–11]. RNA interference (RNAi) is also essential for H3K9 methylation and silencing in heterochromatin[12]. Histone deacetylases (HDACs), including Sir2 and Clr3, mediate the removal of the acetyl group from histones to ensure

[1]State Key Laboratory of Molecular Biology, Key Laboratory of Epigenetic Regulation and Intervention, Shanghai Institute of Biochemistry and Cell Biology, Center for Excellence in Molecular Cell Science, Chinese Academy of Sciences, Shanghai, China. [2]University of Chinese Academy of Sciences, Beijing, China. [3]Department of Biology, New York University, New York, NY, USA. [4]School of Life Science and Technology, ShanghaiTech University, 100 Haike Road, Shanghai, China. [5]Department of Biological Sciences, Faculty of Science, National University of Singapore, Singapore, Singapore. [6]These authors contributed equally: Wenqi Sun, Qianhua Dong, Xueqing Li, Jinxin Gao. ✉e-mail: fl43@nyu.edu; yongchen@sibcb.ac.cn

the hypoacetylation status and are essential for heterochromatin formation[13–15].

As in many other eukaryotic organisms, all heterochromatic regions associate with the nuclear envelope (NE) in interphase in fission yeast[16]. Telomeres in *S. pombe* form one to three clusters on the nuclear membrane[17,18], whereas centromeres are clustered underneath the spindle pole body (SPB), which is localized at the cytoplasmic side of the nuclear envelope[19]. The *mat* locus is also tethered to the NE[20]. Attachment of telomeres and the *mat* locus to the NE is mediated by the inner nuclear membrane (INM) protein Bqt4[17,18]. Bqt4 interacts with the telomeric protein Rap1 to serve as the telomere-attaching anchor on the NE[18]. The attachment of heterochromatin to the NE contributes to heterochromatin silencing. For example, silencing in the *mat* locus is compromised in the *bqt4Δ* mutant[17]. However, the molecular mechanisms by which the spatial organization of heterochromatin ensures epigenetic silencing are still poorly understood.

Liquid-liquid phase separation (LLPS) has recently emerged as an important principle underpinning cellular functions. Proteins with phase separation properties can form liquid-like condensates, which compartmentalize biochemical activities without membranes and thereby enhance the efficiency of cellular processes[21,22]. Phase separation has been implicated in heterochromatin formation[23–26]; however, the role of LLPS in spatial genome organization remains elusive.

Sad1 is a crucial yet less characterized inner nuclear membrane protein. Sad1 belongs to the conserved SUN (Sad1-UNC-84) family proteins and interacts with the KASH-family protein Kms1 to form the linker of nucleoskeleton and cytoskeleton (LINC) complex[16]. Sad1 contains a single transmembrane helix, with the N-terminal domain (NTD) (Sad1$_{NTD}$: amino acids (aa) 1-169) exposed to the nucleoplasm and the rest of the protein localized in the intermembrane lumen space. Sad1 is responsible for tethering centromeres to SPB in vegetative cells[27,28]. It has been shown that the first 60 aa of Sad1 links centromeres with SPB through the centromeric protein Csi1[28,29]. The Sad1$_{NTD}$ also mediates telomere attachment to the NE in meiosis through the meiosis-specific Bqt1/Bqt2 complex[30]. However, the role of Sad1 in heterochromatin localization and silencing in vegetative cells remains unclear.

A previous study showed that the budding yeast SUN protein Mps3 interacts with histone H2A.Z[31]. Here, we demonstrated that Sad1 in fission yeast directly interacts with histone H2A-H2B and H2A.Z-H2B heterodimers, but not H3-H4. We solved the crystal structure of the histone binding motif (HBM) of Sad1 in a complex with a H2A-H2B fusion protein and showed that the DEF/Y motif of Sad1 binds H2A-H2B. To our surprise, we found that in addition to its SPB localization, Sad1 is also dynamically distributed throughout the NE. The NE distribution of Sad1 depends on its interactions with H2A-H2B. We further demonstrated that the Sad1-H2A-H2B complex plays a critical role in the attachment of telomeres and the *mat* locus to the NE. Sad1-H2A-H2B complex is also important for heterochromatin silencing. We identified that Sad1-H2A-H2B interacts with HDACs, including Clr3 and Sir2, to mediate heterochromatin identity. Interestingly, Sad1 exhibits a liquid-liquid phase separation property both in vitro and in vivo. H2A-H2B enhances the phase separation ability of Sad1, which in turn facilitates the recruitment of heterochromatin factors. Our results provide mechanistic insights into the spatial organization of heterochromatin mediated by the conserved SUN-family protein and reveal a previously-unrecognized nucleosome-independent function of free H2A-H2B.

## Results

### Sad1 interacts with histone H2A-H2B

To explore whether the SUN-family protein Sad1 in *S. pombe* has histone binding activity, we first used GST pull-down assays to examine the interaction between the nucleoplasm region of Sad1 (Sad1$_{1-169}$) and

core histone proteins. Sad1$_{1-169}$ could bind the H2A-H2B heterodimer but not the H3-H4 tetramer (Fig. 1a and Supplementary Fig. 1a). Isothermal titration calorimetry (ITC) assays revealed that Sad1$_{1-169}$ could interact with H2A-H2B heterodimer with a dissociation constant ($K_d$) of 7.5 μM and no interaction was detected with H3-H4 (Fig. 1b). The interaction between Sad1 and H2A-H2B was also held when the GST pull-down assay was performed using a single-chain fusion of H2A and H2B (hereafter referred to as H2AB) (Supplementary Fig. 1b), which has been shown to be structurally similar to the wild type H2A-H2B heterodimer[32]. Additionally, Sad1$_{1-169}$ and H2AB could form a stable complex that co-eluted as a single peak in the size exclusion chromatography (Supplementary Fig. 1c).

To determine whether Sad1 could discriminate between H2A and the histone variant H2A.Z, we used GST pull-down assays to compare Sad1$_{1-169}$ binding ability with the H2A-H2B fusion protein (H2AB) and the H2A.Z-H2B fusion protein (H2AZB). GST-Sad1$_{1-169}$ pulled down similar amounts of H2AB and H2AZB (Supplementary Fig. 1b). Isothermal titration calorimetry (ITC) assays further confirmed that Sad1 bound H2AB and H2AZB with comparable dissociation constants ($K_d$) of 9.5 μM and 7.5 μM, respectively (Fig. 1c), indicating that Sad1 does not have a binding preference for H2A or H2A.Z. Therefore, we focus on the interaction of Sad1 with H2A-H2B in the following studies.

To confirm whether Sad1 and H2A-H2B interact with each other in vivo, a fission yeast strain containing HA-tagged Sad1 and FLAG-tagged Htb1 (encoding H2B) at their endogenous genomic sites under native promoters was generated and subject to co-immunoprecipitation (Co-IP) analysis. As a positive and a negative control, we analyzed cells carrying Sad1-HA with either Bqt4-GFP or an inner kinetochore protein Cnp20-TAP, respectively. As expected, Sad1-HA was immunoprecipitated with Bqt4-GFP but not with Cnp20-TAP (Supplementary Fig. 2). Our Co-IP assays using an anti-FLAG antibody clearly showed that Sad1 and H2B co-existed in a protein complex in vivo (Fig. 1d). Collectively, these data reveal that Sad1 has an H2A-H2B binding ability in vivo and in vitro.

### Structural basis for Sad1 interaction with H2AB

To reveal the structural basis of the interaction between Sad1 and H2AB, we first dissected the structural elements of Sad1 required for the Sad1-H2AB interaction. We purified a series of Sad1 fragments in the N-terminal domain to evaluate their interactions with H2AB by GST pull-down assays. A minimal histone binding motif (HBM) of Sad1 (aa 110-140) retained the binding ability with H2AB (Supplementary Figs. 3a,, b). Crystallization screenings were performed by using different Sad1 fragments spanning aa 110-140 of Sad1 in a complex with H2AB. Only the Sad1$_{110-126}$–H2AB complex yielded high-quality crystals that diffracted to 2.15 Å resolution. The crystal structure of Sad1$_{110-126}$-H2AB was solved by molecular replacement using the H2AB fusion protein structure as the search model (PDB: 4WNN)[33], and the crystallographic statistics are presented in Supplementary Table 1. The final model includes H2A 18-127, H2B 30-125, and Sad1 112-123 (Fig. 2a and Supplementary Fig. 4a).

Sad1$_{112-123}$ adopts an extended conformation and primarily contacts H2B (Fig. 2a). The loop between α3 and αC of H2A also contributes to Sad1 binding, and this H2A region is conserved in H2A.Z, explaining why Sad1 does not bear a binding preference for H2A or H2A.Z. The binding between Sad1 and H2AB is mediated by electrostatic and hydrophobic interactions (Figs. 2b, c). The acidic Sad1 peptide sits on the highly basic surface of H2AB (Fig. 2b). A series of acidic residues in Sad1 (D114, E117, E121, and E122) are oriented toward H2AB and form salt bridges and hydrogen bonds with the positively charged residues in H2A and H2B (Fig. 2b and Supplementary Fig. 4b). In addition to electrostatic contacts, the side chain of Sad1 F118 inserts into the shallow hydrophobic groove formed by the α1 helix, L$_{12}$ loop, and α2 helix of H2B, and is embraced by Y42, I54, and M59 of H2B (Fig. 2c), which ensures the specific recognition of H2B by Sad1.

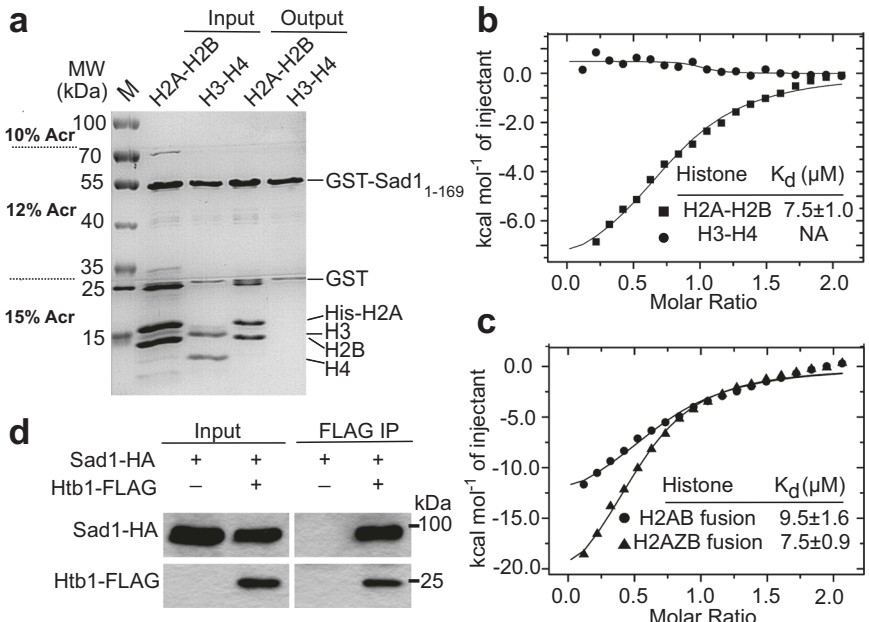

**Fig. 1 | Sad1 interacts with histone H2A-H2B. a** GST pull-down assays showed that GST-Sad1$_{1-169}$ could pull down H2A-H2B heterodimers but not the H3-H4 tetramers. To better separate histone proteins, a home-made segmented gradient gel was used. There are three concentrations of acrylamide from the top to the bottom: 10%, 12%, and 15%. Source data are provided as a Source Data file. **b** Isothermal titration calorimetry analyzes of the interaction between Sad1$_{1-169}$ and H2A-H2B heterodimer (square) or H3-H4 tetramer (dot). The binding isotherms of the titration data and the dissociation constants ($K_d$) are shown. **c** Isothermal titration calorimetry analyzes of the interaction between Sad1$_{1-169}$ and H2AB (dot) or H2AZB (triangle). The binding isotherms of the titration data and the dissociation constants ($K_d$) are shown. **d** Co-immunoprecipitation assays showed that Sad1 and H2B$^{Htb1}$ co-existed in a complex. Lysates from the indicated cells were immunoprecipitated with an antibody specific for FLAG. Immunoprecipitated samples were analyzed by immunoblotting using anti-FLAG and anti-HA antibodies. Source data are provided as a Source Data file.

To corroborate the structural analysis, we purified the proteins with point mutations of the interface residues on Sad1 and H2AB and then used ITC assays to examine whether these mutants could affect the interactions between Sad1$_{60-169}$ and H2AB. The Sad1 F118A mutation destabilized the Sad1-H2AB complex, and the substitution of Sad1 F118 with a positively charged arginine residue (F118R) severely diminished the interaction between Sad1 and H2AB (Fig. 2d). In addition, charge-reverse mutations of D114 and E117 decreased binding affinity between Sad1 and H2AB by 5.6-fold (Fig. 2d). Similarly, the mutations on histones, including H2A R79E and H2B Y42R, also severely reduced the interactions with Sad1 (Fig. 2d). We also created a combined 5 R mutation (F118R/D114R/E117R/E121R/E122R) of Sad1 and found that the mutation completely disrupted its binding with H2AB (Fig. 2d). To examine the effect of 5 R mutation on the interaction between Sad1 and histone in vivo, we created a stain carrying Sad1-5R-HA and H2B$^{Htb1}$-FLAG and conducted Co-IP experiments. Co-IP results demonstrated that Sad1-5R is defective in binding histones in vivo (Supplementary Fig. 4c). Thus, our mutagenesis analyzes supported the importance of hydrophobic and electrostatic interactions in coordinating the interaction between Sad1 and H2AB.

Notably, the core histone-binding fragment of Sad1 contains a DEF/Y motif commonly identified in H2A-H2B or H2A.Z-H2B-binding proteins, including Chz1, Anp32e, Swr1, and the suppressor of Ty16 (Spt16)[33–36] (Fig. 2e), and showed a conserved histone recognition mode (Supplementary Fig. 5). Furthermore, superimposition of Sad1-H2AB with the nucleosome structure showed that the Sad1 peptide occupied the DNA binding site of H2A-H2B in the nucleosome (Fig. 2f), suggesting that Sad1 does not bind nucleosomal H2A-H2B.

## Sad1-H2A-H2B interaction is required for Sad1 distribution on the nuclear envelope

We next investigated the functions of the Sad1-H2A-H2B interaction in vivo. We first examined whether the Sad1-H2A-H2B interaction

contributes to the localization of Sad1. Previous studies have shown that Sad1 associates with SPB, and GFP-tagged Sad1 displays a single bright focus in interphase cells[27,28]. However, after careful examination of cells expressing Sad1-GFP at the endogenous site under its own promoter using fluorescence microscopy, we found that, in addition to a single bright focus at the SPB, weak Sad1-GFP signals were also observed across the NE in more than 82% of cells (Fig. 3a–c). The weak signal forms discrete puncta, the size and number of which vary between different cells (Fig. 3c). The average number of observed Sad1-GFP puncta per cell was 2.9 ± 1.2. The association of weakly-diffused Sad1-GFP with the NE was confirmed by the DeltaVision imaging system using cells with a nuclear envelope marked with mCherry-tagged Ish1 (Fig. 3d). These data indicate that, in addition to its SPB localization, Sad1 is also distributed throughout the NE.

To determine whether the Sad1-H2A-H2B interaction plays a role in Sad1 localization, we created a GFP-tagged Sad1-5R mutant by replacing endogenous *sad1+* with the mutant version by CRISPR. The 5R mutation did not affect the overall structural fold of Sad1, as revealed by the similar gel filtration profiles and thermal melting curves between 5R and WT Sad1 proteins (Supplementary Figs. 6a, b). The mutant cells did not display apparent growth abnormalities. Using the mCherry-tagged SPB protein Sid4 as an SPB marker, we found that the dominant GFP focus of Sad1-5R-GFP colocalized with Sid4-mCherry (Supplementary Fig. S6c), indicating that the Sad1-5R mutation does not affect SPB association of Sad1. However, the signal of Sad1-5R-GFP on the nuclear envelope was lost (Figs. 3a, e). We also replaced endogenous *sad1+* with the GFP-tagged HBM-deleted Sad1, *sad1-ΔHBM-GFP*. We observed that Sad1-ΔHBM-GFP had the same distribution pattern as Sad1-5R-GFP (Fig. 3a–c). These results indicate that the interaction between Sad1 and histones is required for the distribution of Sad1 in the nuclear envelope but not for its localization to the SPB.

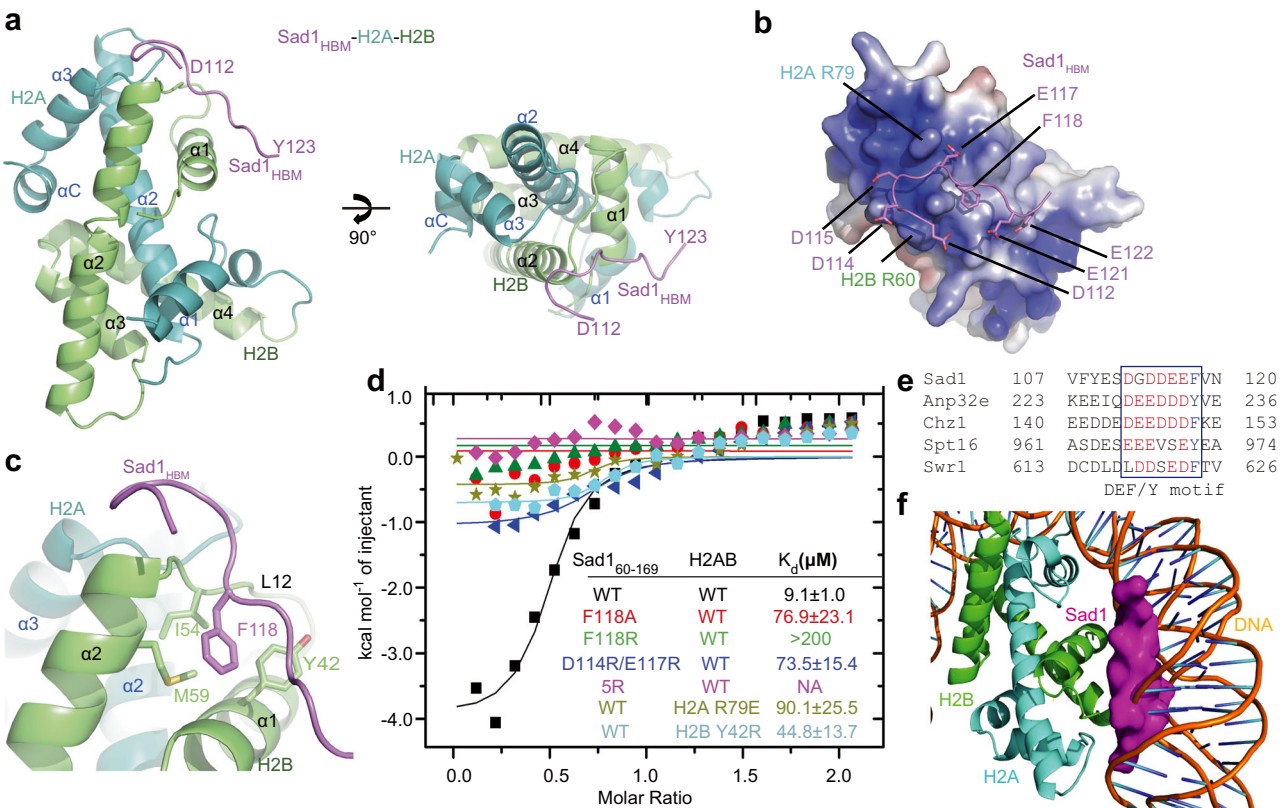

**Fig. 2 | The interface between Sad1_HBM and H2AB. a** The overall structure of the Sad1_HBM-H2AB complex. Sad1_HBM is colored purple, H2A is colored green, and H2B is colored cyan. Two orthogonal views are shown. **b** The electrostatic interface between Sad1_HBM and H2AB. H2AB is shown as the surface model and colored according to its electrostatic potential (positive potential, blue; negative potential, red). **c** Details of hydrophobic interactions between Sad1_HBM and H2AB. The side chain of Sad1 F118 is embraced by Y42, I54, and M59 of H2B. **d** ITC measurements revealed that Sad1 and histone mutations decreased the Sad1-H2AB interaction. Sad1-5R represents Sad1^D114R/E117R/F118R/E121R/E122R. **e** Sequence alignment of the Sad1 DEF/Y motif with other H2A-H2B-binding proteins. The conserved acidic residues and Phe/Tyr residues are highlighted. **f** Superimposition of the Sad1_HBM-H2AB complex structure with the nucleosome structure. The Sad1_HBM peptide clashed with the DNA-binding site of H2A-H2B in the nucleosome.

## The association of telomeres and the mating-type locus with the nuclear envelope depends on Sad1-H2A-H2B

The N-terminus of Sad1 has been shown to be essential for centromere clustering on the nuclear envelope beneath the SPB[28]. We next investigated whether the interaction between Sad1 and histones is important for the centromere-NE association. We created the *sad1-5R* mutant at its native site and examined the distribution of GFP-tagged centromere protein Cnp1 expressed in wild-type (WT) and *sad1-5R* cells. During interphase, WT cells carrying Cnp1-GFP display a single fluorescence focus, indicating centromere clustering at SPB. We found that a single Cnp1-GFP spot was still observed in *sad1-5R* and *sad1-ΔHBM* interphase cells, similar to WT cells (Supplementary Figs. 7a, b). These data indicate that the interaction between Sad1 and histone is not involved in centromere clustering at SPB, which is consistent with the previous finding that Sad1_1-60 is essential for linking centromeres with SPB[28,29].

We next tested whether the histone binding ability of Sad1 is required for telomere tethering to the NE. Telomere position relative to the NE was analyzed in vegetative cells carrying Taz1-GFP and Ish1-mCherry, which were used as markers of telomeres and the NE, respectively. In WT cells, telomeres formed one to three clusters closely associated with the NE (Fig. 4a). By contrast, telomeres were detached from the NE in *sad1-5R* and *sad1-ΔHBM* cells (Fig. 4a). To better quantify the effects of *sad1* mutations on the telomere-NE association, we assigned each Taz1-GFP spot to one of three concentric zones of equal volume with respect to Ish1-mCherry as previously described[37] (Fig. 4b). In WT cells, we found that while most Taz1-GFP

spots were localized in the NE-periphery zone (zone 1), *sad1-5R* and *sad1-ΔHBM* cells had a significantly decreased number of Taz1-GFP spots positioned in zone 1 and an increased number of Taz1-GFP spots in zone 2/3 (Fig. 4c). We further analyzed how the telomere position was affected in *sad1-5R* using a *lacO* array integrated at the *sod2+* locus in the subtelomeric region in cells expressing LacI-GFP. The NE was visualized by Cut11-RFP[17]. This analysis also revealed that telomeres were significantly dissociated from the NE in the mutant (Supplementary Figs. 7c,, d). These data indicate that the histone-binding ability of Sad1 is required for the association of telomeres with the NE.

We also examined the effect of *sad1-5R* on the attachment of the *mat* locus to the NE. The *mat* locus was visualized using the LacI-GFP that binds to the *lacO* repeats array integrated into proximity to the *mat* locus[38]. Using cells carrying the *mat* locus mark and Ish1-mCherry, we also observed that the *mat* locus was dislodged from the NE in *sad1-5R* cells (Figs. 4d, e). These results collectively demonstrate the critical role of the Sad1-histone interaction in the NE attachment of telomeres and the *mat* locus.

## Sad1-H2A-H2B is important for heterochromatin silencing

To determine whether Sad1 on the NE associates with heterochromatin, we examined cells carrying Sad1-GFP and mCherry-Swi6 as a marker for heterochromatin. We found that Sad1-GFP spots on the NE often colocalize with mCherry-Swi6 (Supplementary Fig. 8a). In addition, we performed Chromatin Immunoprecipitation quantitative PCR (ChIP-qPCR) analysis of cells carrying Sad1-GFP with an antibody specific for GFP. We found that Sad1-GFP was highly enriched in telomeres

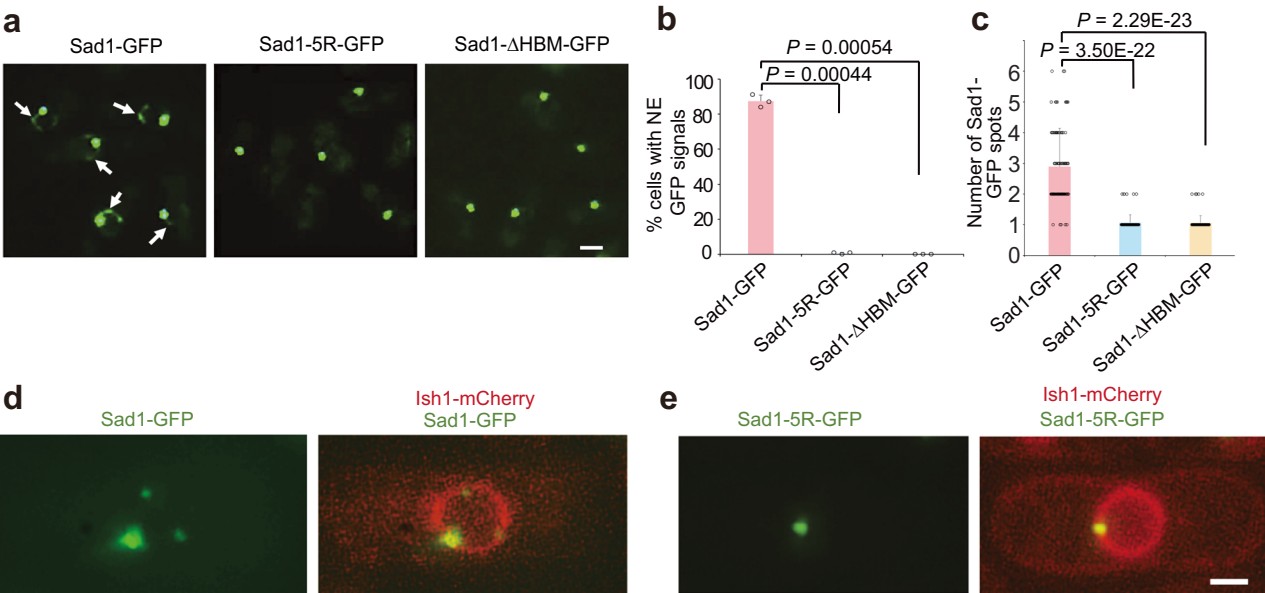

**Fig. 3 | The Sad1-H2A-H2B complex is required for the distribution of Sad1 on the NE. a** Representative images of cells carrying Sad1-GFP, Sad1-5R-GFP, and Sad1-ΔHBM-GFP. Sad1-GFP was present throughout the NE in addition to its association with SPB. The Sad1 mutants (Sad1-5R and Sad1-ΔHBM) lost the NE localization but retained SPB localization. White arrows: Sad1-GFP puncta on the nuclear envelope. Scale bars, 2 μm. **b** Statistical analyzes of the cells with NE-localized GFP signals in (**a**). Three independent experiments were repeated. 100 cells were scored for Sad1-GFP and 120 cells for Sad1-5R-GFP and Sad1-ΔHBM-GFP in one single experiment. Data are presented as mean ± SD. The indicated *P* values are from two-sided

Student's *t* test. Source data are provided as a Source Data file. **c** Statistical analyzes of the total number of fluorescent spots of Sad1-GFP, Sad1-5R-GFP, and Sad1-ΔHBM-GFP on the nuclear envelope per cell. 85 cells were scored for Sad1-GFP, Sad1-5R-GFP, and Sad1-ΔHBM-GFP. Data are presented as mean ± SD. The indicated *P* values are from two-sided Student's *t* test. Source data are provided as a Source Data file. **d** Representative images of yeast cells expressing Sad1-GFP and the nuclear envelope marker Ish1-mCherry. Scale bars, 2 μm. **e** Representative images of yeast cells expressing Sad1-5R-GFP and Ish1-mCherry. Scale bars, 2 μm.

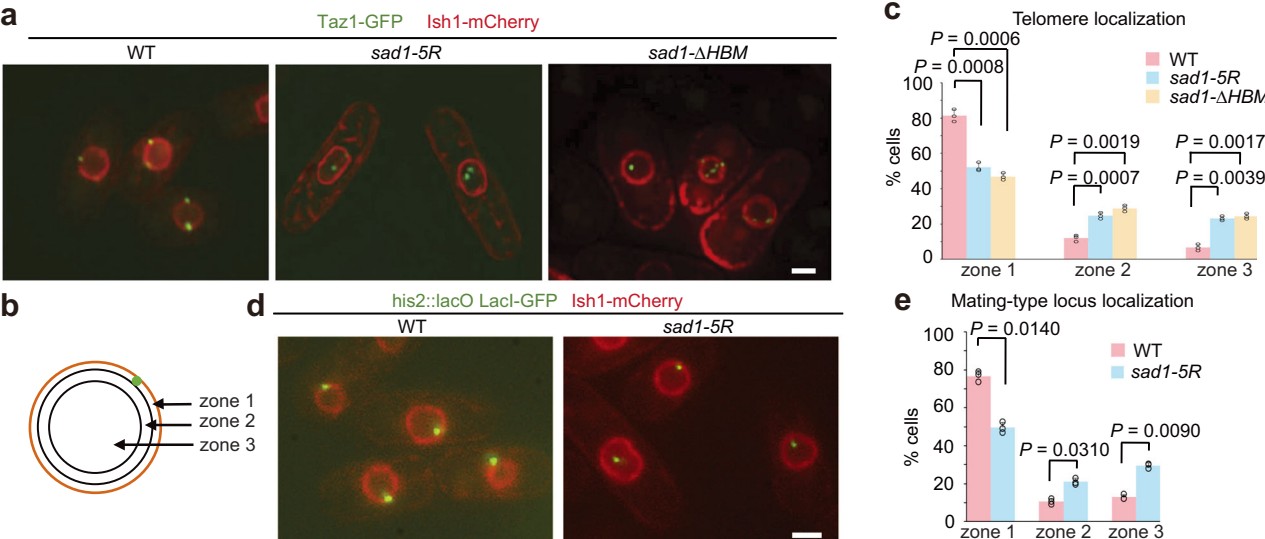

**Fig. 4 | Sad1-H2A-H2B regulates the association of telomeres and the mating-type locus with NE. a** Representative images showing telomere localization in the indicated strains. Telomeres were visualized by Taz1-GFP, and the nuclear envelope was represented by Ish1-mCherry. Scale bars, 2 μm. **b** The subnuclear position of the telomere was scored with respect to the distance from the nuclear envelope and assigned a position in one of the three equal concentric zones of the nucleus. Green, GFP-tagged locus. Orange, nuclear envelope. **c** Percentage of cells with Taz1-GFP positioned in three concentric zones. Three independent experiments were repeated. 200 cells were scored in one single experiment. Data are presented as

mean ± SD. The indicated *P* values are from two-sided Student's *t* test. Source data are provided as a Source Data file. **d** Representative images showed the localization of mating type loci using a mating type locus mark (*his2::lacO-GFP*) in the indicated cells. Scale bars, 2 μm. **e** Percentage of cells with mating-type loci positioned in three concentric zones. Three independent experiments were repeated. 200 cells were scored in one single experiment. Data are presented as mean ± SD. The indicated *P* values are from two-sided Student's *t* test. Source data are provided as a Source Data file.

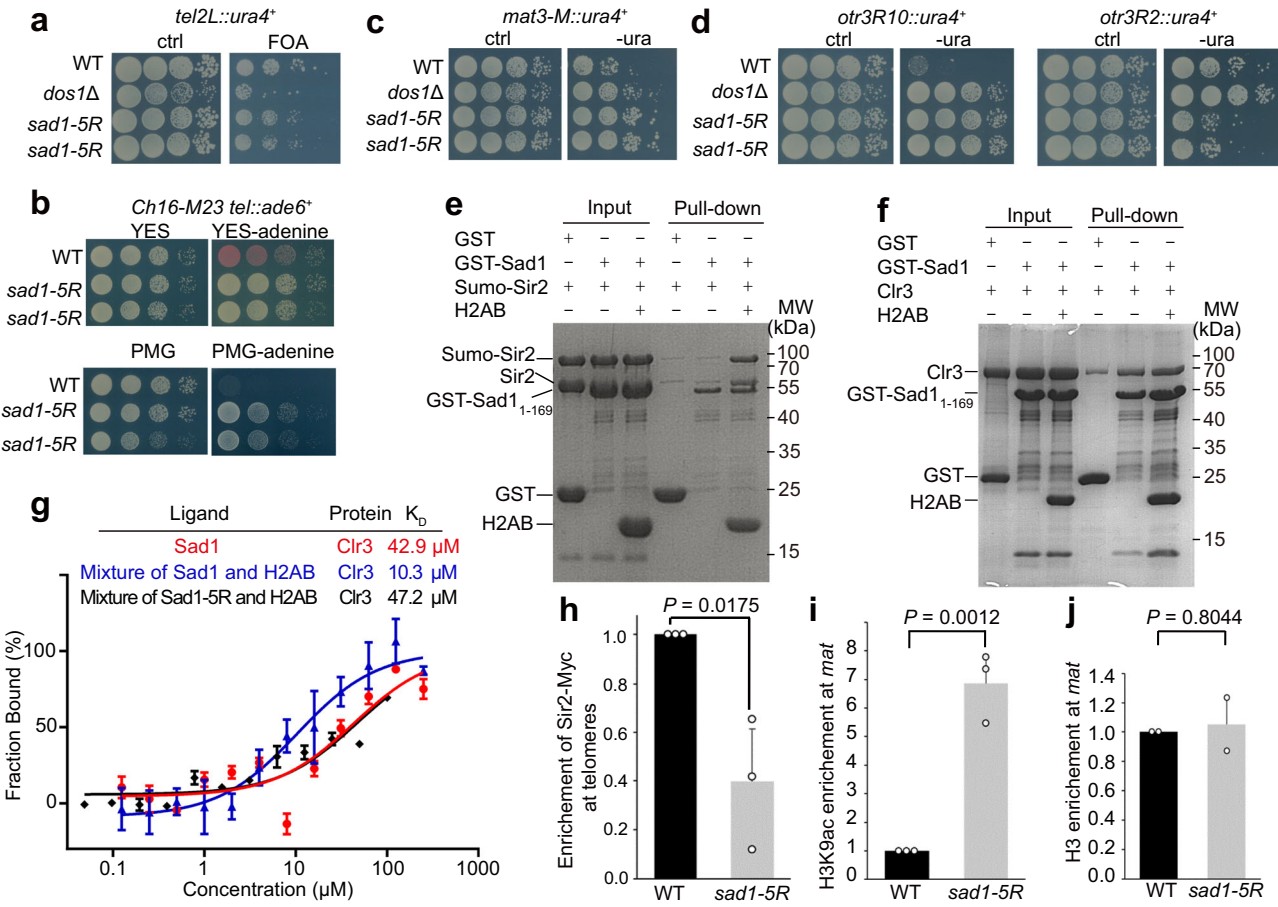

**Fig. 5 | Sad1-H2AB promotes heterochromatin assembly by interacting with Sir2 and Clr3. a** Silencing of telomeres is reduced in *sad1-5R* cells. Serial dilutions of the indicated cells carrying *ura4*⁺ in telomeres (*tel2::ura4*⁺) were plated on 5-FOA counter-selective media and incubated at 30 °C for 4 days. Ctrl. Control. The *dos1Δ* strain served as a positive control in all the silencing assays. Two independent *sad1-5R* colonies were used. **b** Telomere silencing assays using an *ade6*⁺ reporter inserted in telomeres in minichromosome 16. Serial dilutions of the indicated cells were plated on YES low adenine media (YES-ade) (top panel) or PMG-ade (lower panel). The *sad1-5R* mutant resulted in the loss of silencing as shown by colony color (top panel) and growth rate (lower panel). **c** Silencing in the mating-type region is disrupted in *sad1-5R*. Silencing of mating-type loci was tested using an *ura4*⁺ reporter inserted in the mating-type locus. -ura, no uracil. **d** Silencing of the peri-centromeric region was analyzed using the *ura4*⁺ reporter inserted at different *otr3* repeats, including *otr3R2* and *otr3R10*. The *sad1-5R* mutant showed silencing defects in *otr3R10* but not in *otr3R2*. **e** GST pull-down assay showing the interaction between Sad1₁₋₁₆₉ and Sumo-Sir2 in the presence and the absence of H2AB. There is still some untagged Sir2 from self-proteolysis of Sumo-Sir2 overlapped with GST-

Sad1₁₋₁₆₉ on the gel. Source data are provided as a Source Data file. **f** GST pull-down assay showing the interaction between Sad1₁₋₁₆₉ and Clr3 in the presence and the absence of H2AB. Source data are provided as a Source Data file. **g** MST measurements revealed that H2AB could mildly increase the binding affinity between Clr3 and Sad1 (blue line), but not Sad1-5R (black line). Data are presented as mean ± SD from *n* = 3 independent experiments. Source data are provided as a Source Data file. **h** ChIP-qPCR assays showed that Sir2-Myc in the telomeric region in *sad1-5R* was significantly reduced. The level of WT was set to 1. Data are presented as mean ± SD from n = 3 independent experiments. The indicated *P* value is from two-sided Student's *t* test. Source data are provided as a Source Data file. **i** ChIP-qPCR assays showed that H3K9ac in the mating-type locus in *sad1-5R* was significantly increased. Data are presented as mean ± SD from *n* = 3 independent experiments. The indicated *P* value is from two-sided Student's *t* test. Source data are provided as a Source Data file. **j** ChIP-qPCR assays showed that the enrichment of histone H3 in the mating-type locus in *sad1-5R* was similar to WT. Data are presented as mean ± SD from n = 2 independent experiments. The indicated *P* value is from two-sided Student's *t* test. Source data are provided as a Source Data file.

and the *mat* locus (Supplementary Figs. 8b, c), indicating that Sad1 associates with heterochromatin regions.

Next, we wanted to test whether the histone binding activity of Sad1 is important for heterochromatin silencing. We first examined telomere silencing in *sad1-5R* cells. We used a reporter strain with the *ura4*⁺ gene inserted at a subtelomeric region in Chr2. In WT cells, the *ura4*⁺ reporter in the subtelomeric region is transcriptionally silenced, and these cells thus grew well in the media containing 5-fluoroorotic acid (5-FOA), a drug that kills cells expressing *ura4*⁺ (Fig. 5a). We found that *sad1-5R* cells showed slower growth in the 5-FOA media (Fig. 5a), indicating that silencing in subtelomeres is moderately compromised in the Sad1-mutated cells. It should be noted that silencing defect induced by *sad1-5R* was weaker than that in cells lacking Dos1(*dos1Δ*), a

key component of the H3K9 methyltransferase complex[7–10]. To further confirm the role of Sad1-H2A-H2B in telomere silencing, we used a strain that carries the *ade6*⁺ gene inserted in the subtelomere of the minichromosome Ch16-M23. WT cells with the minichromosome exhibit red colonies in low adenine media due to the silencing of *ade6*⁺. By contrast, the *sad1-5R* mutant resulted in the loss of silencing as shown by white colony color in YES-adenine medium and fast growth rate in PMG-adenine medium (Fig. 5b), indicating that subtelomeric silencing in the minichromosome is also partially disrupted.

We also tested heterochromatin silencing at the *mat* locus using strains carrying *ura4*⁺. Our growth assays revealed that the *mat* locus silencing was also lost in the *sad1-5R* mutant (Fig. 5c). We next investigated the effect of *sad1-5R* on peri-centromeric silencing. The

peri-centromeric region in fission yeast contains large outermost repeats (*otr*) as well as innermost repeats (*imr*). We previously showed that the level of silencing in different *otr* repeats varies[39]. Using strains carrying *ura4*⁺ at different *otr3* repeats, we consistently observed that the silencing in Repeat 10 in *otr3* (*otr3R10*) was reduced in *sad1-5R*, but silencing in Repeat 2 in *otr3* (*otr3R2*) was not affected (Fig. 5d). Together, these data demonstrated that the Sad1-H2A-H2B complex is important for silencing in a wide range of heterochromatin regions.

## Sad1-H2A-H2B mediates heterochromatin assembly by recruiting HDACs

We next investigated the molecular mechanism for the Sad1-H2AB-mediated heterochromatin silencing. We surmised that the Sad1-H2A-H2B complex may interact with certain silencing factors to regulate heterochromatin formation. To test this hypothesis, we performed a candidate screen to identify potential silencing factors interacting with Sad1-H2AB using in vitro pull-down assays. By screening some known silencing factors, our in vitro pull-down assays identified that Clr3 and Sir2 directly interact with the Sad1-H2AB complex (Figs. 5e, f), but not Clr2 and Swi6 (Supplementary Figs. 9a, b). Both Clr3 and Sir2 are HDACs important for heterochromatin silencing[13,14]. Notably, in the absence of H2AB, Sir2 could not be pulled down by GST-Sad1$_{1-169}$ (Fig. 5e), and H2AB mostly acts as the bridge to interact with Sir2 and Sad1 simultaneously (Supplementary Fig. 9c). On the contrary, GST pull-down assay did not show strong effect of H2AB enhancing Sad1$_{1-169}$-Clr3 interaction (Fig. 5f). We then used Microscale thermophoresis (MST) assays to quantitatively characterize the interaction between Sad1$_{1-169}$ and Clr3. MST assays showed that the binding affinity between Sad1-H2AB and Clr3 was 3-fold higher than that between Sad1 and Clr3, but the addition of H2AB failed to increase the interaction between Sad1-5R and Clr3 (Fig. 5g). These data suggest that H2A-H2B could enhance the interaction between Sad1 and HDACs, including Sir2 and Clr3.

To further support the role of Sad1 in recruiting HDACs, we used ChIP-qPCR assay to examine how Sad1 affects the enrichment of Sir2 in telomere regions. ChIP-qPCR analysis showed that Sir2-Myc was significantly reduced in telomeres in *sad1-5R* (Fig. 5h). In addition, the double mutants of *sad1-5R* with *clr3Δ* or *sir2Δ* exhibits no synthetic effect in heterochromatin silencing (Supplementary Figs. 9d, e), supporting the idea that Sad1 acts in the same pathway as the HDACs. Consistent with these data, our ChIP-qPCR assays showed that histone acetylation in the *mat* locus in *sad1-5R* was significantly increased (Fig. 5i) while the histone H3 level in the region was similar to WT (Fig. 5j). These results collectively suggest that the Sad1-H2AB interaction cooperatively enhances the interaction between Sad1 and HDACs, which facilitates the recruitment of HDACs to heterochromatin to ensure the correct epigenetic status required for heterochromatin silencing.

## H2A-H2B promotes Sad1 phase separation

We next asked how H2A-H2B and Sad1 orchestrate the interactions with different partners to mediate heterochromatin localization and silencing. The sequence analysis revealed that the Sad1$_{NTD}$ possesses a low complexity sequence (Supplementary Fig. 10a), which could lead to liquid-liquid phase separation (LLPS). Fluorescence microscopy analysis confirmed that purified Sad1$_{1-169}$-GFP fusion proteins could form spherical droplets in the presence of a crowding agent (10% PEG-8000). The droplet size increased with the increasing protein concentrations (Fig. 6a). The formation of Sad1-GFP droplets is dependent on salt concentration: the droplets can be disrupted by higher salt concentrations (Supplementary Figs. 10b, c). The phase separation ability of Sad1 relies on its N-terminal 80 aa, which is distinct from its histone-binding motif (aa 110–140) (Fig. 6b). The Sad1-GFP droplets were near micrometre size and underwent dynamic fusion events (Fig. 6c). Liquid-like condensates are characterized by rapid exchange

kinetics, which can be determined by measuring FRAP (fluorescence recovery after photobleaching). Our FRAP experiments showed that the droplets exhibited fast recovery after photobleaching (Figs. 6d, e). These characteristics observed for the Sad1$_{1-169}$ droplets indicate that Sad1$_{1-169}$ can undergo LLPS in vitro.

Next, we explored whether Sad1 could undergo phase separation in vivo. Sad1-GFP expressed under its own promoter formed distinct puncta on the NE (Figs. 3a, d). Using time-lapse fluorescence microscopy, we found that Sad1-GFP puncta on the NE underwent dynamic movement at a speed of ~0.2 μm minute⁻¹, and also can fuse together (Fig. 6f and Supplementary Movie 1), characteristic of liquid-like condensates. We next used FRAP to examine Sad1-GFP exchange kinetics on the NE in vivo. We first analyzed the cells with mildly overexpressed Sad1-GFP. Sad1-GFP was constructed under the thiamine-repressible *nmt1* promoter and mildly overexpressed in 10 μM thiamine. Sad1-GFP expressed in this condition displayed a similar pattern on the NE as the one under its endogenous promoter but appeared brighter (Fig. 6g). Our FRAP results showed that more than 80% of Sad1-GFP in these cells was recovered within 40 s (Fig. 6h). We also used FRAP to examine cells expressing Sad1-GFP under its native promoter and obtained similar results (Figs. 6i, j). These data thus indicate that Sad1 within the puncta dynamically exchanges with surrounding environments, suggesting that Sad1 has a liquid-liquid phase separation property in vivo.

We then investigated how H2A-H2B binding affects the phase separation behavior of Sad1. The fluorescent H2AB fusion protein (H2AB-RFP) or full-length H2A-H2B heterodimer (H2B + H2A-RFP) also formed liquid-like droplets (Supplementary Figs. 11a, b). Sad1$_{1-169}$-GFP and H2AB-RFP (or H2B + H2A-RFP) can co-exist in the same droplet (Fig. 6k and Supplementary 11c). As controls, purified RFP or GFP alone was not concentrated into Sad1$_{1-169}$-GFP or H2AB-RFP droplets, respectively, and an unrelated protein HP1α-RFP could not be incorporated into Sad1$_{1-169}$-GFP droplets (Supplementary Figs. 11d–f), proving the specific effect of H2A-H2B on the droplet formation of Sad1. Fluorescence images of mixed H2AB-RFP and Sad1$_{1-169}$-GFP proteins showed that H2AB substantially augmented the phase separation of Sad1$_{1-169}$-GFP (Fig. 6k–m). The mixed Sad1$_{1-169}$-GFP and H2AB-RFP form much larger droplets than those from Sad1$_{1-169}$-GFP or H2AB-RFP alone (Figs. 6k, l). Moreover, the inclusion of H2AB increased the salt threshold of Sad1-droplet formation and lowered the critical protein concentration required for Sad1 droplet formation (Fig. 6m and Supplementary 11g).

We further checked the effect of H2AB on the phase separation of the histone-binding-deficient Sad1-5R mutant. Sad1$_{1-169}$−5R itself still retained the similar LLPS ability as WT Sad1 (Supplementary Fig. 12a). However, H2AB-RFP could not be recruited into the Sad1$_{1-169}$−5R-GFP droplets (Fig. 6n), and the size of the droplets remained the same as that of Sad1$_{1-169}$ alone (Fig. 6l). Moreover, it is worth noting that the 147-bp nucleosome core particle (NCP) was not able to promote the phase separation of Sad1 (Supplementary Fig. 12b), suggesting that H2AB-dependent augmentation of Sad1 phase separation relies on its specific interaction with the free H2A-H2B. Thus, Sad1 can undergo liquid-liquid phase separation in vitro and in vivo, and histone H2A-H2B enhances the phase separation ability of Sad1.

## H2A-H2B-enhanced Sad1 phase separation facilitates the recruitment of heterochromatin factors

We hypothesized that H2A-H2B might promote Sad1 phase separation to recruit heterochromatin factors to the same condensates to facilitate heterochromatin organization. Thus, we examined how H2AB-enhanced phase separation of Sad1 affects the recruitment of Clr3 and Sir2 into the Sad1-droplets. Recombinant Clr3-CFP or Sir2-CFP fusion protein alone could not form phase-separated droplets (Figs. 7a, b). The fluorescence images revealed that Clr3-CFP could be incorporated into the Sad1-GFP droplets (Fig. 7a), although Sir2-CFP could not (Fig. 7b). These results are consistent with our pull-down assays

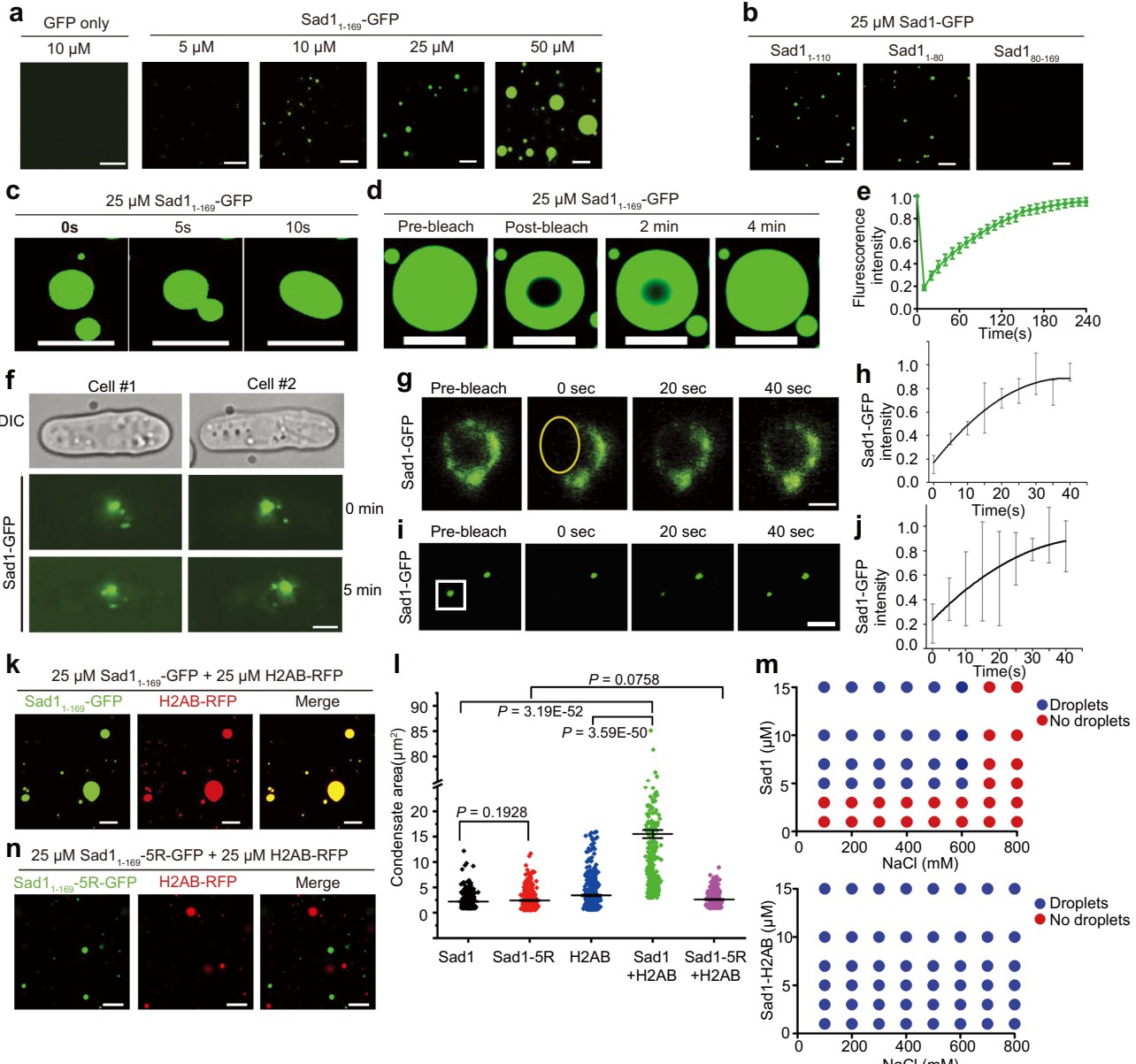

**Fig. 6 | H2A-H2B promotes Sad1 phase separation. a** Fluorescence microscopy images of the Sad1$_{1-169}$-GFP fusion protein at different concentrations in droplet formation buffer containing 150 mM NaCl and 10% PEG-8000. Scale bars, 10 μm. GFP serves as the negative controls and shows no LLPS at the tested concentration. **b** Fluorescence images of different Sad1 constructs in the droplet formation buffer. Scale bars, 10 μm. **c** Fluorescence images of the same view field showing the dynamic fusion of the Sad1 droplets. Scale bars, 10 μm. **d** Fluorescence images of Sad1$_{1-169}$-GFP before and after photobleaching. Scale bars, 10 μm. **e** Quantitation of the normalized fluorescence recovery of Sad1$_{1-169}$-GFP following droplet photobleaching. Data are presented as mean ± SEM from $n = 3$ independent experiments. Source data are provided as a Source Data file. **f** Sad1 exhibits dynamic movement on the nuclear envelope. Time-lapse fluorescence microscopy of Sad1-GFP under its native promoter in WT cells. Scale bar, 2 μm. **g** Fluorescence images of mildly overexpressed Sad1-GFP during a FRAP experiment. The bleached region is indicated with a yellow oval. Scale bars, 1 μm. **h** The plot of normalized recovery of the overexpressed Sad1-GFP signal after photobleaching. Data are presented as mean ± SD from n = 3 independent experiments. Source data are provided as a

Source Data file. **i** Fluorescence images of Sad1-GFP expressed from its native promoter during a FRAP experiment. The bleached region is indicated with a white square. Scale bars, 1 μm. **j** The plot of normalized recovery of the native Sad1-GFP signal after photobleaching. Data are presented as mean ± SD from n = 4 independent experiments. Source data are provided as a Source Data file. **k** Representative images of wild-type Sad1$_{1-169}$-GFP (25 μM) mixed with H2AB-RFP (25 μM). H2AB increased the droplet size of Sad1$_{1-169}$. Scale bars, 10 μm. **l** Quantitation of the condensate area for Sad1$_{1-169}$-GFP (n = 379), Sad1$_{1-169}$−5R-GFP (n = 328), H2AB-RFP (n = 449), Sad1$_{1-169}$-GFP mixed with H2AB-RFP ($n = 371$), and Sad1$_{1-169}$−5R-GFP mixed with H2AB-RFP ($n = 379$). Each protein was 25 μM. Data are presented as mean ± SEM. The indicated $P$ values are from two-sided Student's $t$ test. Source data are provided as a Source Data file. **m** Phase diagrams for LLPS of Sad1 and Sad1-H2AB in the presence of different protein concentrations and NaCl concentrations determined by in vitro LLPS assays. **n** Representative images of Sad1$_{1-169}$−5R-GFP (25 μM) mixed with H2AB-RFP (25 μM) which show no fusion of Sad1$_{1-169}$−5R-GFP and H2AB-RFP droplets. Scale bars, 10 μm.

showing the interaction between Sad1 and Clr3 but no interaction between Sad1 and Sir2 (Figs. 5e, f). Interestingly, in the presence of histone H2AB, in which condition Clr3 and Sir2 can strongly interact with Sad1, both Clr3 and Sir2 could be efficiently concentrated into

Sad1-H2AB droplets, and the sizes of the droplets were substantially increased (Figs. 7a, b).

We next used FRAP to examine Clr3-GFP exchange kinetics in vivo. We found that Clr3-GFP expressed under its native promoter forms

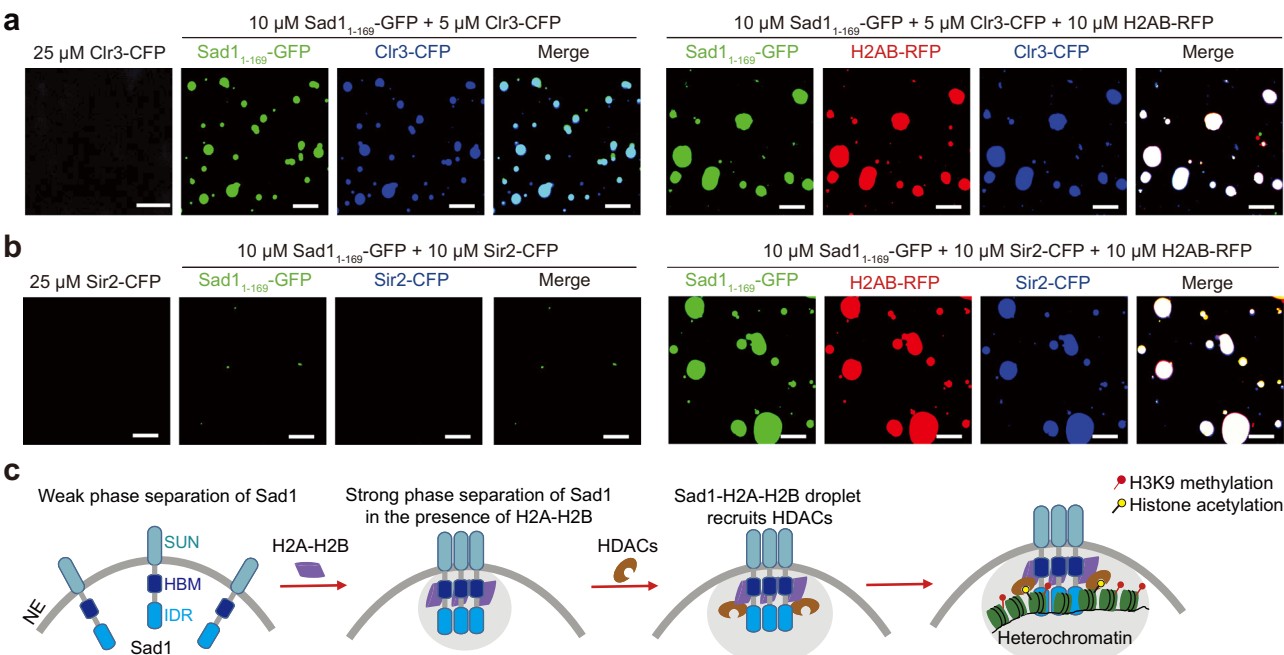

**Fig. 7 | H2AB-enhanced phase separation of Sad1 facilitates HDAC recruitment.**
**a** Fluorescence images of Clr3-CFP alone, Sad1$_{1-169}$-GFP mixed with Clr3-CFP, and a
mixture containing Sad1$_{1-169}$-GFP, H2AB-RFP, and Clr3-CFP. Clr3-CFP alone could
not form liquid-like droplets, but could be incorporated into the same Sad1 dro-
plets and Sad1-H2AB droplets. Scale bars, 10 µm. **b** Fluorescence images of Sir2-CFP
alone, Sad1$_{1-169}$-GFP mixed with Sir2-CFP, and a mixture containing Sad1$_{1-169}$-GFP,
H2AB-RFP, and Sir2-CFP. Sir2-CFP alone could not form liquid-like droplets. Sir2-

CFP could not be incorporated into Sad1 droplets but could be efficiently incor-
porated into Sad1-H2AB droplets. Scale bars, 10 µm. **c** The working model for Sad1's
role in heterochromatin localization and assembly. H2A-H2B-enhanced Sad1 liquid
phase condensation facilitates the recruitment of heterochromatin factors to the
nuclear periphery to promote heterochromatin formation. Losing the H2A-H2B-
binding ability of Sad1 results in defects heterochromatin positioning and
formation.

distinct spots in the nucleus, especially enriched in the NE (Supple-
mentary Fig. 12c), same as previously reported[40]. Our FRAP results
revealed that, similar to Sad1-GFP, most of Clr3-GFP signal was recov-
ered within 40 s after bleaching the Clr3-GFP spots near the NE (Sup-
plementary Fig. 12d), indicating that Clr3 also has a liquid-liquid phase
separation property in vivo. Together, these data demonstrate that
H2A-H2B promotes Sad1 phase separation to enrich Sad1-interaction
partners, such as HDACs, in the same condensates.

Collectively, we propose the following model for Sad1's role in
heterochromatin positioning and formation: H2A-H2B interacts with
Sad1 and facilitates the otherwise-dispersed Sad1 clustering into con-
densates, forming discrete Sad1 puncta distributed on the NE as
observed in Fig. 3; liquid droplet-like condensates formed by Sad1-
H2A-H2B help recruit heterochromatin factors, such as Clr3 and Sir2,
to the condensates, which promotes spatial organization and hetero-
chromatin silencing at the nuclear envelope (Fig. 7c).

## Discussion
In this study, we revealed that the conserved SUN-domain protein Sad1
together with free histone H2A-H2B serves as a previously unrecog-
nized regulatory module to mediate heterochromatin organization
near the nuclear periphery (Fig. 7c). Our studies uncovered crucial
roles of the SUN-domain protein Sad1 in heterochromatin regulation.
First, Sad1 associates with heterochromatin regions, including cen-
tromeres, telomeres, and mating-type locus (Supplementary
Fig. 8a–c)[28]. Second, Sad1 is important for heterochromatin spatial
localization to the NE. Previous studies showed that the N-terminal 60
aa of Sad1 is involved in centromere clustering at SPB[28]. Here, we show
that the attachment of telomeres and the *mat* locus to the NE relies on
the histone binding motif of Sad1 (aa 110-140). Thus, Sad1 uses two
motifs to regulate spatial positioning of different heterochromatin
regions on the NE. Third, Sad1, together with H2A-H2B, is important for

heterochromatin silencing. Heterochromatin silencing is compro-
mised in subtelomeres and the *mat* locus as well as some pericen-
tromeric region in the *sad1-SR* mutant. Mechanistically, Sad1 interacts
with HDACs, including Clr3 and Sir2, in an H2A-H2B-dependent man-
ner to ensure the hypoacetylation status required for heterochromatin
identity[13,14]. Moreover, our data suggest that Sad1 has a histone-
enhanced LLPS property that can help concentrate factors required for
efficient heterochromatin organization, reinforcing the prevailing
paradigm that LLPS of silencing factors drives heterochromatin
formation[23,25,26,41]. These unique properties of Sad1 collectively provide
mechanistic insights into Sad1-mediated heterochromatin organiza-
tion. How Sad1-H2A-H2B specifically targets to heterochromatins and
whether other factors (proteins or RNA) are involved merit further
investigation.

The elucidated role of Sad1 in heterochromatin regulation is
reminiscent of recent findings that some inner nuclear membrane
proteins are critical heterochromatin regulators. The nuclear mem-
brane complex, Lem2-Nur1, is involved in heterochromatin localiza-
tion and assembly[42–44]. Another nuclear membrane protein, Amo1, also
promotes the positioning and silencing of the *mat* locus[45]. We hypo-
thesize that these nuclear membrane proteins may form a synergistic
network to create a microenvironment to benefit heterochromatin
organization and maintenance at the nuclear periphery, which requires
further investigation.

Fission yeast Sad1 binds both canonical histone H2A-H2B and
histone variant H2A.Z-H2B, whereas the budding yeast SUN-family
protein, Mps3, specifically recognizes H2A.Z[31]. Similar to the Sad1-H2A-
H2B interaction, the interaction between Mps3 and H2A.Z-H2B is also
required for the distribution of Mps3 on the nuclear envelope[31].
However, the interaction between Mps3 and H2A.Z is not necessary for
telomere-NE association or heterochromatin silencing[31,46], although
Mps3 itself is required for telomere anchoring at the nuclear periphery

and heterochromatin silencing in telomeres[47]. The difference between Sad1 and Mps3 may reflect the distinct epigenetic environments in these species. Fission yeast contains more complex heterochromatin with epigenetic components, such as H3K9me and RNAi, which are missing in budding yeast. We speculate that these differences may lead to the different behavior of Sad1 and Mps3, suggesting the evolutionary plasticity of SUN-family proteins.

Histones are the most abundant nuclear proteins and act as the constitutive components of chromatin. As the building blocks of chromatin, the majority of previous studies have focused on their nucleosome-dependent functions. Whether nuclear free histones have any nucleosome-independent role remains elusive. Our results revealed a nucleosome-independent function of H2A-H2B: H2A-H2B physically associates with Sad1 to regulate Sad1's interaction with other factors, such as Sir2 and Clr3, and promote the LLPS ability of Sad1. Thus, in this case, H2A-H2B functions as a modulator or "chaperone" for Sad1 to enhance Sad1's functionality. Recently, free H2A-H2B has also been identified as a bona fide component of telomerase holoenzymes and could stabilize the folding of telomerase RNA[48,49]. Our results and recent studies highlight the intriguing notion that free H2A-H2B could have broader nucleosome-independent functions than previously thought.

## Methods

### Protein expression and purification

All Sad1 (Uniprot: Q09825) fragments and mutations were cloned into a pGEX-6P-1 vector with an N-terminal GST tag. The *S. pombe* H2B/H2A or H2B/H2A-RFP heterodimer was obtained by separately cloning H2B (Uniprot: P04913) and H2A (Uniprot: P04909) (or H2A-RFP) into the first and second promoters of the modified pCDF-Duet vector, respectively. The fusion proteins of *S. pombe* H2AB (H2B$_{32-126}$-H2A$_{15-108}$) and H2AZB (H2B$_{32-126}$-H2A.Z$_{22-115}$) were cloned into the first promoter of a modified pCDF-Duet vector with an N-terminal 6*His tag followed by a 3 C protease site. EGFP (abbreviated as GFP, Genbank: ACC91771.1), ECFP (abbreviated as CFP, Genbank: AFV14779.1), MRFP (abbreviated as RFP, Genbank: AEA29852.1), Clr3 (Uniprot: P56523), Clr2 (Uniprot: O13881), Sir2 (Uniprot: O94640), Swi6 (Uniprot: P40381), Sad1-GFP, H2AB-RFP, Clr3-CFP, Sir2-CFP, and HP1α-RFP (P45973) were inserted into the modified pET28b vector with an N-terminal 6*His-Sumo tag. All constructs were expressed in *Escherichia coli* Rosetta cells. For protein expression, the cultures were induced for 16–18 h with 0.2 mM IPTG at 16 °C. Cells were harvested after induction and suspended in lysis buffer (50 mM Tris-HCl pH 8.0, 400 mM NaCl, 10% glycerol, protease inhibitor cocktail, and 2 mM β-mercaptoethanol). For histone purification, lysis buffer with 2 M NaCl was used. Cells were disrupted by sonication and centrifuged at 40,000 g for 50 min. The supernatant was loaded onto glutathione Sepharose 4B beads (GE Healthcare, USA) for GST-tag fused proteins or incubated with Ni-NTA agarose beads (Qiagen, Germany) for 6*His-Sumo tag fused proteins at 4 °C for 2 h. After extensive washing with lysis buffer, target proteins were digested by 3C (for pGEX-6P vector or pCDF-Duet vector) or Ulp1 (for pET28b-vector) protease on beads. If GST-tagged and Sumo-tagged proteins were required (e.g., proteins used in GST pull-down assays), proteins were eluted with 15 mM reduced glutathione or 300 mM imidazole. The eluted proteins were further purified with a Hiload Superdex 75 or 200 column (GE Healthcare, USA). Fractions were analyzed with SDS-PAGE, and purified proteins were then concentrated and stored at −80 °C.

### Crystallization, data collection, and structure determination

*S. pombe* H2AB fusion protein (H2B$_{32-126}$-H2A$_{15-108}$) and Sad1$_{110-126}$ were mixed at a molar ratio of 1:2 on ice for 30 min, and then the mixture was used for crystallization. The complex was crystallized in 0.2 M potassium nitrate and 20% w/v PEG-3350. The diffraction data were then collected at the beamline BL19U1 of the Shanghai Synchrotron

Radiation Facility (SSRF). Data were processed using HKL3000[50]. Structure determination was achieved by molecular replacement in Phaser using the yeast H2A-H2B structure (PDB:4WNN) as the initial model[33,51]. Structural refinement was accomplished in the PHENIX package[52]. Model refinement and building were performed in COOT[53].

### GST pull down assays

Recombinant GST-tagged Sad1$_{1-169}$ and fused H2AB or H2AZB were mixed in 50 μl binding buffer (25 mM Tris-HCl, pH 8.0, 400 mM NaCl, and 2 mM DTT). Then, 10 μl glutathione Sepharose 4B beads (GE Healthcare, USA) were added and incubated at 4 °C for 1 h. After the beads were extensively washed three times with 200 μl binding buffer, the remaining bound proteins were eluted with 40 μl elution buffer (15 mM reduced GSH, 400 mM NaCl, 25 mM Tris-HCl, pH 8.0, and 2 mM DTT). The eluted protein was detected by SDS-PAGE and stained with Coomassie blue. To test Sad1 interactions with HDACs, the same GST pull-down assays were performed by using GST-tagged Sad1$_{1-169}$. For pull-down assays to test GST-Sad1$_{1-169}$ interaction with Sir2, the Sumo-Sir2 fusion protein was used because the untagged Sir2 runs at the similar position as GST-Sad1$_{1-169}$ on the gel. For pull-down assays to test GST-Sad1$_{1-169}$ interaction with Clr3, the untagged Clr3 was used.

### Isothermal titration calorimetry

Proteins used for ITC measurements were dialyzed against the assay buffer (25 mM Tris-HCl, pH 8.0, 400 mM NaCl, and 2 mM DTT). ITC titrations were conducted on a MicroCal ITC200 system (Malvern Panalytical Ltd, UK) at 20 °C, by which a high concentration of Sad1 (500 μM) was titrated into H2AB (50 μM). Raw data were subsequently analyzed and plotted with Origin 7 software (OriginLab, USA). All ITC titration experiments were performed at least in duplicate, and one representative curve was shown.

### Microscale thermophoresis (MST) assay

All MST measurements were performed on a Monolith NT.115 system (Nano Temper Technologies, Germany). For MST measurements, all samples were dialyzed into the same MST assay buffer containing 25 mM HEPES, pH 7.5, 150 mM NaCl, and 2 mM DTT. Clr3 was labeled with the EZLabel Protein FITC Labeling Kit (BioVision, USA). FITC labeled Clr3 was diluted in MST buffer to 20 nM. Ligands (500 μM Sad1$_{1-169}$ or 250 μM Sad1$_{1-169}$-H2AB complex) were diluted by 12 gradients. Then the ligands and the labeled Clr3 were mixed and incubated in the darkroom for 20 min. Capillaries were then filled with samples individually and loaded into the instrument with the following thermophoresis parameters: medium MST power and 80% LED. MST data were processed with MO Control Software (Nano Temper Technologies, Germany), and the binding affinity was determined with MO Affinity Analysis software (Nano Temper Technologies, Germany).

### Nano differential scanning fluorimetry (NanoDSF) assay

NanoDSF measurements were carried out on a Prometheus NT.48 system (NanoTemper Technologies, Germany). Protein samples were diluted to 1 mg/ml in the buffer containing 25 mM Tris-HCl, pH 8.0, and 150 mM NaCl. Capillaries were then filled with samples individually and loaded into the instrument and heated from 20 °C to 85 °C at the heating rate of 1 °C/min. The intrinsic fluorescence emission at 330 nm and 350 nm was recorded. The melting temperature(Tm) was determined by the PR. ThermControl software (NanoTemper Technologies, Germany).

### In vitro phase separation assays

The phase separation assays were performed as previously described[54]. The proteins and DNA used for droplet formation were adjusted to the desired concentrations in the phase separation buffer composed of 25 mM Tris-HCl, pH 8.0, 150 mM NaCl, and 10% PEG-8000. Then 3–6 μl of the mixed sample was deposited onto a

microscope slide. Imaging of the droplets was captured by a Zeiss LSM 710 microscope (Zeiss, Germany).

The fluorescence recovery after photobleaching (FRAP) experiments was performed on a Zeiss LSM 710 microscope with a 63x oil objective, following the procedure previously described[55]. In our in vitro FRAP experiment, the region of interest (ROI) (diameters of 4 to 8 μm) was bleached for 300 iterations using a laser with 488 nm wavelength at normal 100% laser transmission. Then the post-bleach time-lapse images were collected every 10 s for 5 min. The mean fluorescence intensities from the photobleached region of every time-lapse image were measured and recorded. The raw data were processed and plotted by GraphPad Prism.

### Strains, media, and genetic analysis
Standard media and genetic analysis for fission yeast were used[56]. The *sad1* mutants were created by CRISPR/Cas9-mediated gene editing as previously described[57]. For all spot tests, cells were grown to log phase (~1 × 10^7 cells/ml), and then 10-fold serial dilutions were spotted onto the indicated media and incubated at 30 °C for 3 to 5 days. Fission yeast strains used in this study are listed in Supplementary Table 2.

### Western Blot Analysis
Cell extracts from exponentially growing cells were prepared using standard protocols and a bead beater. Extracted proteins were separated on 10-15% SDS-polyacrylamide gels and blotted onto PVDF membranes. Standard western blot protocols were used in subsequent steps.

### Immunoprecipitation
Immunoprecipitation was performed as previously described[58] with minor modifications. Cells were grown in YES rich media at 30 °C until the mid-log phase. Cell extracts were incubated with the GFP-Trap magnetic agarose (gta-20, Chromotek, Germany), anti-FLAG (F1804, Sigma, USA), or anti-GFP HRP (ab190584, Abcam, UK) at 4 °C for 1 h. After washing with lysis buffer three times, proteins were eluted in SDS loading buffer. Eluates were analyzed by Western blotting using commercial anti-GFP HRP (ab190584, Abcam, UK), anti-FLAG antibody (F1804, Sigma, USA), anti-HA antibody (12CA5, Roche, USA), anti-TAP antibody (P1291, Sigma, USA), and goat anti-mouse IgG H&L (HRP) secondary antibody (ab6789, Abcam, UK).

### ChIP-qPCR
ChIP was performed as previously described[59]. 100 mL log phase cells were fixed with 1% formaldehyde (F8775, Sigma, USA) for 30 mins at room temperature. The cell wall was digested with Zymolase 20 T at 30 °C for 30 mins. Cell lysates were incubated with anti-GFP antibody (ab290, Abcam, UK) and Protein A agarose beads (223-50-00, KPL, USA). The precipitated DNA was suspended in 20 μl TE buffer. 2 μl of ChIP or WCE (whole cell extract) samples were analyzed by qPCR (4367659, Applied Biosystems, USA) with primers specific to the mating type region (mal-F: GAAAACACATCGTTGTCTTCAGAG; mal-R: TCGTCTTGTAGCTGCATGTGA), sub-telomere region (jk380: TAT TTCTTTATTCAACTTACCGCACTTC; jk381: CAGTAGTGCAGTGTATTA TGATAATTAAAATGG), and the control gene, *act1* + (Act1: ATGGAA-GAAGAAATCGCAGCG; Act2: GATGCCAAATCTTTTCCATATC).

### Microscopy
Cells were imaged using a BX53 fluorescence microscope (Olympus, Japan) or the Delta Vision System (Applied Precision, USA). For the Delta Vision System, images were taken as z-stacks of 0.2 μm increments with an oil immersion objective (100x) and deconvolved using SoftWoRX2.50 software (Applied Precision, USA). The method for analyzing the positioning of telomeres and the mating-type locus within the nuclear zones was conducted as described[37]. The signals were quantified using ImageJ v1.53a (NIH, USA) as described

previously[60]. To capture time-lapse images, cells were adhered to an agarose pad on slides as described previously[61], and time-lapse intervals are indicated in the corresponding figure legends.

### In vivo FRAP assays
The cellular FRAP experiments were performed using the Zeiss Air-yscan Confocal system (Zeiss, Germany). Three laser lines of the Argon laser (458, 488, 514 nm) were used to bleach the GFP signals in the areas of interest at 100% transmission and 10 iterations. The bleach was started after one acquisition scan, and the images were acquired at the indicated intervals.

### Statistics and reproducibility
The experiments of pull-down assay and Co-IP assays were performed at least twice and similar results were obtained, so the representative results were shown in Figs. 1a, d, 5e, f, Supplementary Fig. 1a, b, 2a, b, 3b, 4c, 9a–c. All ITC experiments were repeated twice, and one representative plot was shown in Figs. 1b, c, and 2d. All the fluorescence images for in vitro LLPS assays were taken from at least 10 different regions and the representative images were shown in Figs. 6a–c, k, n, 7a, b, Supplementary Figs. 10b, 11a–g, 12a, b. The fluorescence images in Figs. 3d, e, Supplementary Figs. 6c, and 8a are representative images from at least 60 different yeast cells which showed similar image patterns. Data were analyzed using an unpaired two-sided student *t*-test, and *P* values < 0.05 were considered statistically significant.

### Reporting summary
Further information on research design is available in the Nature Portfolio Reporting Summary linked to this article.

## Data availability
Coordinates and structure factors of Sad1_HBM-H2AB has been deposited in the Protein Data Bank under accession codes 7YBF. The following structures were used in the paper for structural analysis: 6AE8, 4WNN, 4CAY, 4M6B. Source data are provided with this paper. All the protein constructs are available from Dr. Yong Chen, and all the yeast strains are available from Dr. Fei Li upon request. Source data are provided with this paper.

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

## Acknowledgements

We thank staff members of the BL19U1 beamline of the National Facility for Protein Science Shanghai (NFPS) at Shanghai Synchrotron Radiation Facility for X-ray diffraction data collection. We are grateful to the staff members of the mass spectrometry system, the nuclear magnetic resonance system, and the large-scale protein production system of NFPS for providing technical support and assistance in data collection and analysis. We also thank S. Broyde for critical reading of the manuscript and S. Chen for assistance in FRAP and strain construction. We thank J.P. Cooper and the Japan Yeast Genetic Resource Center for the strains. This work was supported by grants from the Strategic Priority Research Program of the Chinese Academy of Sciences (XDB37010303 to Y.C.), the Shanghai Pilot Program for Basic Research – Chinese Academy of Science, Shanghai Branch (JCYJ-SHFY-2022-008 to Y.C.), the National Natural Science Foundation of China (31970576 to Y.C.), the NIH grant (R35GM134920-01 to F.L.) and the NSF grant (MCB-1934628 to F.L.).

## Author contributions

Y.C. and F.L. conceived and supervised the project. W.S., X.L. and X.Y. purified the proteins and performed the biochemical assays. W.S. and C.H. performed crystallization and determined the crystal structure. Q.D. and J.G. performed the in vivo studies. W.S., Q.D., X.L., J.G., F.L. and Y.C. prepared the figures and wrote the manuscript.

## Competing interests

The authors declare no competing interests.
