## [Peer Review File · Nature Communications]

The SUN-family protein Sad1 mediates heterochromatin spatial organization through interaction with histone H2A-H2BEditorial Note: Parts of this Peer Review File have been redacted as indicated to remove third-party material where no permission to publish could be obtained.

REVIEWER COMMENTS

Reviewer #1 (Remarks to the Author):

The *S. pombe* Sad1 protein, a member of the SUN-domain family, is a nuclear membrane protein. Sad1 is known to mediate the attachment of centromeres and telomeres to the nuclear membrane in vegetative and meiotic cells, respectively. In this work, Chen and colleagues report the crystal structure of Sad1 complexed with the H2A-H2B dimer and demonstrated the biological significance of the interaction through numerous biochemical and cell biological studies. The authors show that the histone-binding activity of Sad1 is essential for tethering telomeres and the mating-type locus to the nuclear membrane and for heterochromatin silencing. The authors also demonstrate that Sad1 interacts with histone deacetylases, Clr3 and Sir2, critical factors involved in heterochromatin silencing. Sad1 also exhibits liquid-liquid phase separation property, which the H2A-H2B dimer enhances. The authors conclude that Sad1 regulates heterochromatin function by interacting with the H2A-H2B dimer in a nucleosome-independent manner, an unexpected role for the SUN-domain nuclear membrane proteins.

The manuscript is well-written and easy to follow. For the most part, the results are clear, and the interpretation of the data is warranted. I have a few concerns that should be addressed before publication in Nature Communications. Please consider the following comments.

Major points:

1) I am unconvinced that the interaction between Sad1 and H2A-H2B is nucleosome-independent. From the model (Figure 7C), the authors propose that Sad1 interacts with free H2A-H2B not incorporated into the nucleosome. They reason that "the Sad1 peptide occupied the DNA binding site of H2A-H2B in the nucleosome" (p. 8, lines 162-163). However, this DNA binding site of H2B corresponds to the SHL 5~6 location of the nucleosomal DNA, which is close to the entry/exit site. Several nucleosome structures are reported in which the nucleosomal DNA ends are unwrapped from the histone core. Thus, it is conceivable that Sad1 can bind to nucleosomes in which the DNA is partially unwrapped.

2) Figure 1A

a) The molecular mass of GST-Sad1(1-169) is approximately 45 kDa but appears to be 55 kDa in the gel image. This region of Sad1 is acidic, so the charge does not seem to cause an increase in the apparent molecular mass. Please clarify this. Similarly, the molecular masses of the fragments shown in Figure S3 are also inconsistent with the molecular weight marker.

b) I needed clarification on why GST alone is observed. Is this a result of partial proteolysis of the GST-Sad1 sample? Furthermore, the migration distances in the four lanes are different. This is awkward because they all should be the same sample. The authors may wish to redo the electrophoresis and/or provide explanations. In addition, the migration distances of the molecular weight markers are quite different from that shown in Figures S1 and S3 (in particular, the 35 kDa marker band).

3) p. 9, lines 183-184

"We still observed a dominant GFP focus associated with SPB in the mutant." Is this statement true? The authors should also visualize SPB with a different fluorescent protein and see if SPB co-localizes with the Sad1 mutant.

4) Figures 5E and 5F

a) Please show molecular weight markers.

b) In Figure 5E, Sir2 and GST-Sad1(1-169) in the input are not separated. The authors may wish to redo the electrophoresis (perhaps with lesser amounts of protein or with a different gel concentration).

c) In Materials and methods, Clr3 appears to be expressed with a SUMO tag, but in Figure 5F, Clr3 seems free of the tag. Please clarify this.

5) Figure 7C

In the bottom three models, Sad1 appears to associate with each other during phase separation freely. However, the C terminal side of Sad1 is embedded in the inner nuclear membrane, so their movements are likely more restricted than proteins that can freely move in the nuclear lumen. The authors should modify the model so that it takes into consideration the membrane association of Sad1.

Reviewer #2 (Remarks to the Author):

Nature Communications #426048

The SUN-family protein Sad1 mediates heterochromatin 1 spatial organization through interaction with histone H2A-H2B

In this manuscript, the authors identified an interaction between a nuclear envelope protein, Sad1 and non-nucleosomal H2A-H2B, but not H3-H4. A minimal histone binding domain (110-140 aa) was identified for Sad1. They performed crystal structure and several other studies to reveal the Sad1-H2A-H2B interaction details and showed that such interaction is required for Sad1's nuclear envelope distribution, the tethering of telomeres and the mating-type locus to nuclear envelope, and heterochromatin silencing. They suggest that the role of H2A-H2B here is to enhance the interaction between Sad1 and HDACs, in which a liquid phase separation property of Sad1 is observed. This is an interesting report and has some potential on our understanding of heterochromatin repression mechanisms at the nuclear envelope region. The experimental design is logic, the manuscript is clearly written, the data are convincing and generally support the conclusion. However, the story is rather premature, lacks detailed mechanisms, and therefore current results cannot support the proposed model.

Major concerns:

1. The biggest issue is the unclarity as to how free H2A-H2B facilitated the heterochromatin localization of Sad1 to NE. Superimposition showed that Sad1 blocked DNA binding of H2A-H2B, ruling out the possibility of Sad1 in binding to nucleosomal H2A-H2B. Then how could free H2A-H2B proteins guide Sad1 to heterochromatic DNA in the nuclear peripheral region? If Sad1 prevented H2A-H2B from binding to DNA, how can these proteins maintain stable complex formation at heterochromatic DNA to suppress the transcriptional repression?
2. Another major issue is the lack of causative link between Sad1 phase separation and heterochromatin function. Whether the phase separation feature of Sad1, alone or together with other HDACs, indeed affects the heterochromatin silencing is completely unclear. There is no demonstration of endogenous Sad1 proteins co-phase separate with HDACs at the NE, and the lack of evidence showing that loss of Sad1 phase separation disrupts HDACs' heterochromatin location and transcriptional silencing. Can a HBM intact but phase separation deficient Sad1 still locate to heterochromatin but not inhibit heterochromatin repression? In all, there is no functional connection between the phase separation property and the function of Sad1-H2A-H2B interaction.
3. Lastly, there is no demonstration of functional significance of the two HDACs, Clr3 and Sir2, in the role of Sad1 at heterochromatin. Only complex formation was shown.

Other concerns:

1. It is unclear if Sad1 binds individual H2A and H2B.
2. How close is the crystal structure of the H2A-B fusion protein to natural H2A-H2B heterodimer? Is it possible that the preference of Sad1-HBM contacting with H2B, but not H2A, comes from the H2A-B fusion protein?
3. Related to Fig. 3, there is not data to show that the NE Sad1 signals indeed co-localize with telomeres and the mating locus.
4. Color difference in Fig. 5B is not clearly visible.

5. Fig. 5E-F show that while Sad1 interaction with Sir2 depends on H2A-B, that of Cl3 is not. Will Sad1-5R not interact with Clr3 or Sir2?
6. The rigor of the in vivo FRAP data (Fig. 6) is lacking. How many events were evaluated in Fig. 6E, 6H and 6J? The data also lack statistical analysis. The FRAP result in Fig. 6J apparently does not match the images in Fig. 6I where very weak recovery was shown.

Reviewer #3 (Remarks to the Author):

In eukaryotes, genomic DNA is known to be heterochromatinized and silenced at the nuclear periphery, but the molecular mechanisms underlying heterochromatin organization at the nuclear periphery remains elusive. In this study, the authors investigated the interaction between an inner nuclear membrane protein, Sad1, and the histone H2A-H2B dimer. Mutational studies suggested that the interaction between Sad1 and H2A-H2B was essential for telomere tethering at the nuclear envelope. Additionally, the authors demonstrated that this interaction plays a crucial role in heterochromatin formation and droplet formation. These results indicate a novel regulation mechanism of heterochromatin organization with Sad1 and H2A-H2B dimer. However, some data are preliminary and the mechanism by which the interaction between Sad1 and free H2A-H2B dimer is involved in chromatin organization is still unclear. A significant revision of the work is required, including carefully addressing the major points below.

Major comments:

Comment 1: The pull-down assay in Figure 1A lacks the control experiment with GST and the His-H2A-H2B dimer. The authors should show that the His-H2A-H2B dimer does not bind non-specifically to GS4B beads or GST.

Comment 2: The ITC experiments in Figures 1 and 2 suggest a stoichiometry of 1:2 between H2AB and Sad1. However, the crystal structure reveals a 1:1 complex formation between H2AB and Sad1. The authors should provide an explanation for this discrepancy.

Comment 3: In Figure 4, the authors indicate telomere dissociation from the nuclear envelope with the Sad1 5R mutant by observing Taz1 loci. However, the Sad1 5R mutant may affect the Taz1 interaction with telomere without effects on heterochromatin tethering at the nuclear envelope. The authors should provide direct evidence that heterochromatin region dissociates from the nuclear envelope with the Sad1 5R mutant.

Comment 4: I find it intriguing that Sad1 appears to be required for the association of telomeres with the nuclear envelope, as shown in Figure 4, whereas in Figure 5 the effect of Sad1 on telomere organization at the nuclear envelope appears to be modest. The authors should comment on these two results, to satisfy the reader. The authors should also prove the reproducibility of their ura4⁺ reporter assays shown in Figures 5 A and B, to confirm the effect of the Sad1 5R mutant.

Comment 5: In Figure 6G, the authors did not conduct FRAP analysis on large puncta, such as those shown in the bottom panel of Figure 6F or Figure 6I. I would suggest to the authors that performing photobleaching on a larger area would make the quantification more precise and easier.

Comment 6: There appears to be a discrepancy between the quantitative plot in Figure 6J and the fluorescence recovery after photobleaching shown in Figure 6I. The authors should provide an explanation for this discrepancy or repeat the quantification.

Comment 7: Error bars should be added to Figures 6E, 6H, and 6J to represent statistical significance in figures.

REVIEWER COMMENTS

Reviewer #1 (Remarks to the Author):

The *S. pombe* Sad1 protein, a member of the SUN-domain family, is a nuclear membrane protein. Sad1 is known to mediate the attachment of centromeres and telomeres to the nuclear membrane in vegetative and meiotic cells, respectively. In this work, Chen and colleagues report the crystal structure of Sad1 complexed with the H2A-H2B dimer and demonstrated the biological significance of the interaction through numerous biochemical and cell biological studies. The authors show that the histone-binding activity of Sad1 is essential for tethering telomeres and the mating-type locus to the nuclear membrane and for heterochromatin silencing. The authors also demonstrate that Sad1 interacts with histone deacetylases, Clr3 and Sir2, critical factors involved in heterochromatin silencing. Sad1 also exhibits liquid-liquid phase separation property, which the H2A-H2B dimer enhances. The authors conclude that Sad1 regulates heterochromatin function by interacting with the H2A-H2B dimer in a nucleosome-independent manner, an unexpected role for the SUN-domain nuclear membrane proteins.

The manuscript is well-written and easy to follow. For the most part, the results are clear, and the interpretation of the data is warranted. I have a few concerns that should be addressed before publication in Nature Communications. Please consider the following comments.

We thank this reviewer for the appreciation of our work and constructive suggestions.

Major points:

1) I am unconvinced that the interaction between Sad1 and H2A-H2B is nucleosome-independent. From the model (Figure 7C), the authors propose that Sad1 interacts with free H2A-H2B not incorporated into the nucleosome. They reason that "the Sad1 peptide occupied the DNA binding site of H2A-H2B in the nucleosome" (p. 8, lines 162-163). However, this DNA binding site of H2B corresponds to the SHL 5~6 location of the nucleosomal DNA, which is close to the entry/exit site. Several nucleosome structures are reported in which the nucleosomal DNA ends are unwrapped from the histone core. Thus, it is conceivable that Sad1 can bind to nucleosomes in which the DNA is partially unwrapped.

We thank this reviewer for raising this point. To address whether Sad could bind the nucleosome, we prepared the nucleosome with 147-bp widow-601 sequence and checked Sad1-nucleosome interaction by EMSA assay. As shown in the reference figure 1, Sad1 cannot bind the nucleosome at the test condition. Another evidence is from the inability of the nucleosome to enhance the LLPS ability of Sad1 (Reference Fig. 1B), which is in sharp contrast with the stimulation effect of H2AB on Sad1 LLPS. These results collectively suggest that Sad1 does not bind the nucleosome.

We also analyzed whether H2A-H2B in a partially unwrapped nucleosome could be structurally compatible with Sad1 binding. We used the spontaneously unwrapped nucleosome (PDB: 6ESH, SHL6-7 unwrapped) for structural comparison (Bilokapic, Strauss et al. 2018). When we superimposed the crystal structure of Sad1-H2AB with the unwrapped nucleosome structure (PDB: 6ESH), it clearly shows that Sad1 still clashes with nucleosomal DNA SHL5-6 (Reference Fig. 1C). To avoid clash, the nucleosomal DNA SHL 4.5-6 has to be unwrapped to enable Sad1 binding. However, such a two-turn unwrapping of nucleosomal DNA will destabilize the nucleosome, leading to the dissociation of one H2A-H2B dimer to generate hexasome. Thus, Sad1 cannot bind nucleosomal H2A-H2B.

Reference Figure 1. Sad1 cannot bind the nucleosome.

- A. EMSA assays showed that Sad1 cannot shift nucleosome bands. The nucleosome is 147-bp nucleosome at 1 μM concentration. The Sad1 concentrations range from 1 to 128 μM.
- B. The addition of nucleosomes could not promote Sad1 phase separation.
- C. Superimposition of the Sad1_{HBM}-H2AB complex structure with the unwrapped nucleosome structure (PDB: 6ESH). The Sad1_{HBM} peptide (cyan) shown in the surface model clashed with the DNA-binding site of H2A-H2B in the nucleosome.

2) Figure 1A

a) The molecular mass of GST-Sad1(1-169) is approximately 45 kDa but appears to be 55 kDa in the gel image. This region of Sad1 is acidic, so the charge does not seem to cause an increase in the apparent molecular mass. Please clarify this. Similarly, the molecular masses of the fragments shown in Figure S3 are also inconsistent with the molecular weight marker.

We are sorry about the confusion. DNA sequencing confirmed that we used the correct construct. However, in SDS-PAGE, Sad1_{N169} indeed runs at the much-larger position than the nominal molecular weight. Sad1_{N169} is around 19 kDa, but untagged Sad1_{N169} runs at 35 kDa and GST-Sad1_{N169} runs at 55 kDa (Reference Fig. 2a). The abnormal migration behavior of Sad1 in SDS-PAGE was also observed in some other Sad1 constructs as shown in Figure S3. We noticed that a recent publication (London, Medina-Pritchard et al. 2023), which studies Sad1-Mis18 interaction, also showed that

the band of purified Sad1_{N169} is between 28 and 38 KDa in SDS-PAGE (Reference Fig. 2b), confirming this abnormal migration behavior of Sad1.

Reference Figure 2. Sad1 runs at a much larger position in SDS-PAGE than its nominal molecular weight.

a. SDS-PAGE for a typical GST-Sad1_{N169} purification. The GST-Sad1_{N169} was on-beads digested on the glutathione sepharose 4B beads. The flow through after on-beads digestion (labeled as Eluted Sad1_{N169}) was further purified through size exclusion chromatography (HiLoad 200). GST-Sad1_{N169} runs at 55 KDa, and untagged Sad1_{N169} runs at 35KDa.

b. The recently published paper also showed that the band of Sad1_{N169} is between 28 and 38KDa. This panel is cited from reference (London, Medina-Pritchard et al. 2023) Figure 4B.

b) I needed clarification on why GST alone is observed. Is this a result of partial proteolysis of the GST-Sad1 sample? Furthermore, the migration distances in the four lanes are different. This is awkward because they all should be the same sample. The authors may wish to redo the electrophoresis and/or provide explanations. In addition, the migration distances of the molecular weight markers are quite different from that shown in Figures S1 and S3 (in particular, the 35 kDa marker band).

We thank you for raising this issue. During the GST-fusion protein purification process, GST-Sad1 gradually lost the GST tag due to the partial proteolysis. We tried several strategies to avoid the self proteolysis, but it was inevitable.

We are sorry for the confusion about the different band migration behaviors in Figure 1, S1, and S3. We used different types of gels in these figures to better separate different protein bands. For example, in Figure 1, because histone proteins have very close molecular weights and are relatively small, we used a home-made segmented gradient gel (top part: 10% acrylamide; middle: 12% acrylamide; bottom: 15% acrylamide) (Reference Fig. 3). In this kind of gel, the histone proteins can be well separated. For Figure S1 and S3, because we used a single H2A-H2B fusion protein and didn't need to separate different histone proteins, we used a regular 12% acrylamide gel. In the revision, we have clearly stated the gel system used in the figure legends.

Reference Figure 3. The original figure 1a was performed in a home-made segmented gradient gel. There are three concentrations of acrylamide from the top to the bottom: 10%, 12%, and 15%. The boundary between each acrylamide concentration was clearly visible in the gel as indicated by the arrows.

3) p. 9, lines 183-184

"We still observed a dominant GFP focus associated with SPB in the mutant." Is this statement true? The authors should also visualize SPB with a different fluorescent protein order and see if SPB co-localizes with the Sad1 mutant.

Thanks for the reviewer's suggestion. As the reviewer suggested, in the revised manuscript, we used a SPB marker, Sid4-mCherry to label the SPB. New Supplementary Fig. S6c showed that Sad1-5R-GFP focus indeed colocalizes with SPB marker Sid4-mCherry, confirming that 5R mutation does not affect the SPB localization of Sad1.

4) Figures 5E and 5F

a) Please show molecular weight markers.

Thanks. The molecular weight markers now have been included in the revised Figs. 5e and 5f.

b) In Figure 5E, Sir2 and GST-Sad1(1-169) in the input are not separated. The authors may wish to redo the electrophoresis (perhaps with lesser amounts of protein or with a different gel concentration).

c) In Materials and methods, Clr3 appears to be expressed with a SUMO tag, but in Figure 5F, Clr3 seems free of the tag. Please clarify this.

Here we address points b) and c) together. Both Clr3 and Sir2 are expressed as SUMO-fused protein, and SUMO tag can be removed by ULP1. We used ULP1-digested tag-free Clr3 in the pull down assay as shown in Figure 5F. Initially, we also tried to use tag-free Sir2 in the Sir2 pull down assay, but the tag-free Sir2 was very close to GST-Sad1_{N169} and they could not be well separated. Thus, in Figure 5E, we used the Sumo-Sir2 fusion protein instead of untagged Sir2 in this pull-down assay (Fig. 5e). Sumo-Sir2 and GST-Sad1 can be well separated, clearly supporting the observation that GST-Sad1 can efficiently pull down Sumo-Sir2 only in the presence

of H2AB (Fig. 5e). However, the purified sumo-Sir2 itself unfortunately has inevitable partial proteolysis, thus containing some untagged Sir2 overlapped with GST-Sad1. We have clearly stated the protein samples we used in the revised figure legend.

5) Figure 7C

In the bottom three models, Sad1 appears to associate with each other during phase separation freely. However, the C terminal side of Sad1 is embedded in the inner nuclear membrane, so their movements are likely more restricted than proteins that can freely move in the nuclear lumen. The authors should modify the model so that it takes into consideration the membrane association of Sad1.

Thank you for the suggestion. We only considered the nuclear portion of Sad1 (Sad1_{N169}) in the original version of the model. We have modified Figure 7c to better show the membrane association of Sad1.

Reviewer #2 (Remarks to the Author):

Nature Communications #426048

The SUN-family protein Sad1 mediates heterochromatin 1 spatial organization through interaction with histone H2A-H2B

In this manuscript, the authors identified an interaction between a nuclear envelope protein, Sad1 and non-nucleosomal H2A-H2B, but not H3-H4. A minimal histone binding domain (110-140 aa) was identified for Sad1. They performed crystal structure and several other studies to reveal the Sad1-H2A-H2B interaction details and showed that such interaction is required for Sad1's nuclear envelope distribution, the tethering of telomeres and the mating-type locus to nuclear envelope, and heterochromatin silencing. They suggest that the role of H2A-H2B here is to enhance the interaction between Sad1 and HDACs, in which a liquid phase separation property of Sad1 is observed. This is an interesting report and has some potential on our understanding of heterochromatin repression mechanisms at the nuclear envelope region. The experimental design is logic, the manuscript is clearly written, the data are convincing and generally support the conclusion. However, the story is rather premature, lacks detailed mechanisms, and therefore current results cannot support the proposed model.

We thank this reviewer's critical and constructive comments, which greatly help us to improve the manuscript.

Major concerns:

1. The biggest issue is the unclearness as to how free H2A-H2B facilitated the heterochromatin localization of Sad1 to NE. Superimposition showed that Sad1 blocked DNA binding of H2A-H2B, ruling out the possibility of Sad1 in binding to nucleosomal H2A-H2B. Then how could free H2A-H2B proteins guide Sad1 to heterochromatic DNA in the nuclear peripheral region? If Sad1 prevented H2A-H2B from binding to DNA, how can these proteins maintain stable complex formation at heterochromatic DNA to suppress the transcriptional repression?

Thanks for these comments. We want to emphasize that we revealed a nucleosome-independent role of free H2A-H2B in the present study. Histones are the most abundant nuclear proteins. As the building blocks of chromatin, the majority of previous studies have focused on their nucleosome-dependent functions. Besides nucleosome histones, there are some free histones in cells. Whether free histones have any nucleosome-independent role remain elusive. Recently, free H2A-H2B has also been identified as a bona fide component of telomerase holoenzymes and it has been suggested that free H2A-H2B could stabilize the folding of telomerase RNA (Ghanim, Fountain et al. 2021, Wan, Ding et al. 2021). In the current study, we reveal that free H2A-H2B, but not nucleosomal H2A-H2B, functions as a modulator for Sad1 to enhance Sad1's functionality: First, H2A-H2B binding enhances the interaction between Sad1 and HDACs; Second, H2A-H2B enhances the LLPS ability of Sad1,

which can more efficiently recruit HDACs into Sad1 condensates. Together, NE-anchored Sad1 and free H2A-H2B serve as a regulatory hub to facilitate the recruitment of HDAC to the nuclear periphery to ensure the correct epigenetic states in the heterochromatin regions.

In the revision, we have performed new experiments to demonstrate that Sad1 indeed associates with heterochromatin. We found that Sad1-GFP spots on the NE often colocalize with mCherry-Swi6, a standard heterochromatin marker (Supplementary Fig. S8a). In addition, we performed ChIP-qPCR analysis of cells carrying Sad1-GFP with an antibody specific for GFP. We found that Sad1-GFP was highly enriched in telomeres and the *mat* locus (Supplementary Fig. S8b and S8c), indicating that Sad1 associates with heterochromatin regions. However, as for “how could free H2A-H2B proteins guide Sad1 to heterochromatic DNA in the nuclear peripheral region?”, we cannot address how the Sad1-histone complex specifically targets to heterochromatin at this stage. It may require additional factors (proteins or RNAs). We think it is beyond the scope of this study; we have acknowledged this point in the Discussion session (Page 17).

2. Another major issue is the lack of causative link between Sad1 phase separation and heterochromatin function. Whether the phase separation feature of Sad1, alone or together with other HDACs, indeed affects the heterochromatin silencing is completely unclear. There is no demonstration of endogenous Sad1 proteins co-phase separate with HDACs at the NE, and the lack of evidence showing that loss of Sad1 phase separation disrupts HDACs' heterochromatin location and transcriptional silencing. Can a HBM intact but phase separation deficient Sad1 still locate to heterochromatin but not inhibit heterochromatin repression? In all, there is no functional connection between the phase separation property and the function of Sad1-H2A-H2B interaction.

Thanks for the reviewer's comments. We would like to first clarify that in our manuscript, we proposed that the histone-enhanced Sad1 phase separation, not Sad1 phase separation per se, regulates heterochromatin function. This is supported by our data, including: 1) histones enhance Sad1 LLPS property in vitro, and mutation in the histone binding domain of Sad1 leads to significant loss of LLPS in vitro (Fig. 6n-l); 2) Sad1 also showed LLPS property in vivo (Fig. 6f-j); mutation in the histone binding domain of Sad1 leads to loss of Sad1-GFP on the NE in vivo (Fig. 3a); 3) Sad1-HDAC interactions are strongly enhanced by free H2A-H2B histone (Fig. 5e-g); 4) H2A-H2B-dependent Sad1-HDAC interaction facilitates HDAC incorporation into Sad1 condensates (Fig. 7a-b); 5) Sad1 mutant losing histone-binding ability results in severe heterochromatin defects (Figs. 4 and 5). We apologized for being unclear, and revised our manuscript to clarify this point (Pages 2, 5, 15, 17, and 41).

To further address the reviewer's concern about the relationship between Sad1 phase separation and heterochromatin function, in the revised manuscript, we present

new data showing that Sad1 on the NE associates with heterochromatin regions by fluorescence imaging and ChIP-qPCR (Supplementary Fig. S8a-c). Since Clr3 is known to associate with all heterochromatin regions (Sugiyama, Cam et al. 2007), Sad1 thus colocalizes with Clr3 in heterochromatin. Our new data shows that Clr3 can also form liquid-like condensates on the NE in vivo (Supplementary Fig. 12c-d). Moreover, our additional new data demonstrated that Sir2 in heterochromatin region is significantly reduced in the *sad1-5R* mutant (Fig. 5h); more importantly, the double mutants of *sad1-5R* with *clr3* Δ or *sir2* Δ exhibits no synthetic effect in heterochromatin silencing (Supplementary Fig. S9d and S9e), supporting the idea that Sad1-H2A-H2B acts in the same pathway as the HDACs.

Based on our data, we proposed that histone-enhanced Sad1 phase separation recruits HDACs to heterochromatin to maintain the correct heterochromatin states. That is supported by our data that Sad1 directly interacts with Sir2 and Clr3 (Figs. 5e-g), and Sad1-H2A-H2B interaction facilitates the recruitment of HDACs to Sad1 condensates in vitro (Figs 7a and 7b). In addition, the histone binding-defective *sad1* mutants show significant silencing defects in heterochromatin region (Figs. 5a-5d), decreased HDAC enrichment in heterochromatin (Fig. 5h), and loss of histone acetylation level in heterochromatin (Fig. 5i). Together, these data support that endogenous Sad1 proteins co-phase separate with HDACs at the NE to regulate the heterochromatin function.

We also thank this reviewer's suggestion to use "an HBM intact but phase separation deficient Sad1 still locate to heterochromatin but not inhibit heterochromatin repression". It would be a good idea to disrupt Sad1 phase separation by removing the intrinsically disordered domain (1-80) in Sad1 and to see how the heterochromatin is affected, as the reviewer suggested. Our data showed that the N-terminal region (1-80) of Sad1 is important for phase separation (Fig. 6b). However, this region (1-80) is essential for centromere-NE association (Fernandez-Alvarez, Bez et al. 2016) and also is involved in the interaction with other proteins, e.g. Mis18 complex, (London, Medina-Pritchard et al. 2023). Thus, it prevents us from creating the LLPS-defective mutant simply by truncating this region. We have to carefully identify the essential LLPS-driven point mutations while not affecting other Sad1 functions or interactions, which needs a lot of more work; despite this, it is still possible that we cannot isolate this kind of viable mutants. In addition, we also believe that it is beyond the scope of this manuscript. Nevertheless, as mentioned above, our data strongly support our model that histone-enhanced Sad1 phase separation regulates heterochromatin function.

3. Lastly, there is no demonstration of functional significance of the two HDACs, Clr3 and Sir2, in the role of Sad1 at heterochromatin. Only complex formation was shown.

Thanks for this constructive suggestion. To address the reviewer's concern, we performed the ChIP-qPCR analysis and found that the association of Sir2 with heterochromatin is significantly reduced (new Fig. 5h). This is consistent with our

model that Sad1 facilitates the recruitment of the HDACs to heterochromatin. We also created the double mutants, *sad1-5R sir2Δ* and *sad1-5R clr3Δ*. Our data showed that the double mutants did not show synthetic defects in heterochromatin silencing (new Supplementary Fig. S9d and S9e), consistent with the idea that Sad1 acts in the HDAC pathway.

Other concerns:

1. It is unclear if Sad1 binds individual H2A and H2B.

Thank you for this comment. Because individual H2A or H2B are insoluble when expressed by itself, we could not use GST pull-down or ITC assays to test Sad1 interaction with individual H2A or H2B. However, the structural and biochemical analyses (Fig. 2a-2d) show that both H2A and H2B are required for the interaction. Since the single mutation on H2A or H2B decreased the interaction with Sad1 (Fig. 2d), we believe Sad1 cannot bind individual H2A or H2B.

2. How close is the crystal structure of the H2A-B fusion protein to natural H2A-H2B heterodimer? Is it possible that the preference of Sad1-HBM contacting with H2B, but not H2A, comes from the H2A-B fusion protein?

Thank you for the comments. The H2AB fusion protein has been widely used in previous histone chaperone studies (Hong, Feng et al. 2014, Liang, Shan et al. 2016, Wang, Liu et al. 2019, Huang, Sun et al. 2020) and has been shown to have an identical structure to the natural H2A-H2B heterodimer (Warren, Bonanno et al. 2020), demonstrating that the fusion protein is structurally and biochemically similar to the wild type H2A-H2B heterodimer.

We also want to point out that although Sad1 primarily contacts H2B, a loop between $\alpha 3$ and αC of H2A also contributes to Sad1 binding (Fig. 2a). H2A R79 from this loop mediates the electrostatic interaction with Sad1 (Fig. 2b). Mutation of H2A R79 severely decreased Sad1-H2AB interaction (Fig. 2d). We have modified our text to more precisely describe the interface between Sad1 and H2AB (Page 7).

3. Related to Fig. 3, there is not data to show that the NE Sad1 signals indeed co-localize with telomeres and the mating locus.

We thank the reviewer for the helpful suggestions. In the revision, we created a new strain containing Sad1-GFP and Swi6-mCherry. Swi6, a human HP1 homolog, associates with heterochromatin regions and serves as the heterochromatin marker. During interphase, centromeres are clustered into one spot, while telomeres are clustered to 1-4 spots and the mating type region is close to the centromeres. We thus usually observe 2-5 Swi6-mCherry foci associated with the NE. Our new data showed that Sad1-GFP spots typically colocalize with Swi6-mCherry (Supplementary Fig. S8a). In addition, we performed ChIP-qPCR analysis of cells carrying Sad1-GFP with an antibody specific for GFP. We found that Sad1-GFP was highly enriched in telomeres and the *mat* locus (Supplementary Figs. S8b and S8c), indicating that Sad1 indeed co-localized with telomeres and mating locus.

4. Color difference in Fig. 5B is not clearly visible.

Thanks for the suggestion. According to the reviewer's request, a new experiment has been done and the new data is presented in the updated Figure 5b. To better indicate the silencing loss, in addition to the color change in YES-Ade medium, we also used PMG-ade medium to examine cell growth. The *sad1-5R* mutant resulted in the loss of telomere silencing in minichromosome 16 as shown by colony color (top panel) and growth rate (lower panel) (Fig. 5b).

5. Fig. 5E-F show that while Sad1 interaction with Sir2 depends on H2A-B, that of Clr3 is not. Will Sad1-5R not interact with Clr3 or Sir2?

Thank you for the suggestion. As the reviewer pointed out, the GST pull-down assay in Figure 5E-F shows that the Sad1-Sir2 interaction depends on H2A-H2B, but did not show a strong effect on enhancing the interaction between Sad1 and Clr3. Considering GST pull-down is not a quantitative assay and may not be able to detect the mild difference in Sad1-Clr3 in the presence or absence of histones, we thus used MST (MicroScale Thermophoresis) to quantitatively characterize the interaction between Sad1 and Clr3. MST showed that H2A-H2B enhanced the Sad1-Clr3 interaction by 3-fold. According to this reviewer's suggestion, in the revision, we also characterized the interaction between Sad1-5R and Clr3. As expected, H2A-H2B cannot enhance the interaction between Sad1-5R and Clr3 (Fig. 5g).

6. The rigor of the in vivo FRAP data (Fig. 6) is lacking. How many events were evaluated in Fig. 6E, 6H and 6J? The data also lack statistical analysis. The FRAP result in Fig. 6J apparently does not match the images in Fig. 6I where very weak recovery was shown.

Thanks for the suggestions. We repeated the FRAP experiments at least three times, and showed the statistical analysis in the revised figures (Figs. 6e, 6h, and 6j).

In Fig 6i, Sad1-GFP was expressed at an endogenous level. The pre-beach Sad1-GFP signal on the NE is already weak compared to the overexpressed signal in Fig 6g (Note: Fig 6g and Fig 6i do not have the same contrast). Please note that taking a picture every 5 seconds during FRAP is a kind of photo-bleaching (although less intense), so this weak Sad1-GFP signal is subject to decay due to multiple photo-bleaching during the FRAP process. Therefore, the recovered signal after 40s are very weak compared to pre-bleach. The kind of imaging-induced intensity loss has significant effect on the weak fluorescence spots as seen in Fig. 6i, but the strong fluorescence spots are less affected as seen in Fig. 6d and 6g. On the other hand, the fluorescence recovery curve shown in our figures were normalized, thus appearing not to exactly match the image in Fig. 6i. Similar results have also been shown previously (Ebrahimi, Masuda et al. 2018).

Reviewer #3 (Remarks to the Author):

In eukaryotes, genomic DNA is known to be heterochromatinized and silenced at the nuclear periphery, but the molecular mechanisms underlying heterochromatin organization at the nuclear periphery remains elusive. In this study, the authors investigated the interaction between an inner nuclear membrane protein, Sad1, and the histone H2A-H2B dimer. Mutational studies suggested that the interaction between Sad1 and H2A-H2B was essential for telomere tethering at the nuclear envelope. Additionally, the authors demonstrated that this interaction plays a crucial role in heterochromatin formation and droplet formation. These results indicate a novel regulation mechanism of heterochromatin organization with Sad1 and H2A-H2B dimer. However, some data are preliminary and the mechanism by which the interaction between Sad1 and free H2A-H2B dimer is involved in chromatin organization is still unclear. A significant revision of the work is required, including carefully addressing the major points below.

We appreciate thoughtful comments and constructive suggestion from this reviewer.

Major comments:

Comment 1: The pull-down assay in Figure 1A lacks the control experiment with GST and the His-H2A-H2B dimer. The authors should show that the His-H2A-H2B dimer does not bind non-specifically to GS4B beads or GST.

Thank you for the suggestion. In supplementary Figure S1a, we now provide the new data showing that although His-H2A-H2B has some non-specific binding to beads, GST-Sad1_{N169} could pull down much more His-H2A-H2B than GST.

Comment 2: The ITC experiments in Figures 1 and 2 suggest a stoichiometry of 1:2 between H2AB and Sad1. However, the crystal structure reveals a 1:1 complex formation between H2AB and Sad1. The authors should provide an explanation for this discrepancy.

Thank you for this comment. Please note that the molar ratio when binding is saturated in ITC does not represent the stoichiometry (Reference Fig. 4a). Here we listed the original ITC figure for Sad1 titrated into the H2A-H2B heterodimer (Reference Fig. 4b). The N value is around 0.8. According to our practical experience, due to the potential offsets between actual concentration and measured concentration, we may treat the binding between Sad1 and H2A-H2B as 1:1, consistent with the crystal structure.

[REDACTED]

Reference Figure 4. Isothermal Titration Calorimetry (ITC) assay for detecting Sad1-histone interactions.

- A representative ITC curve shows how the binding affinity, stoichiometry, and enthalpy of the binding reaction can be obtained. The figure is cited from the ITC manufacturer website (<https://www.malvernpanalytical.com/en/products/technology/microcalorimetry/isothermal-titration-calorimetry>).
- The ITC assay showed the binding of Sad1 to H2A-H2B heterodimer. The upper panel is the heat change upon titration of Sad1 into H2A-H2B, and the lower panel is the binding isotherm profile. The obtained binding affinity, stoichiometry, entropy, and enthalpy were shown in the insert.

Comment 3: In Figure 4, the authors indicate telomere dissociation from the nuclear envelope with the Sad1 5R mutant by observing Taz1 loci. However, the Sad1 5R mutant may affect the Taz1 interaction with telomere without effects on heterochromatin tethering at the nuclear envelope. The authors should provide direct evidence that heterochromatin region dissociates from the nuclear envelope with the Sad1 5R mutant.

Thanks for the suggestion. To address the reviewer's concern, we instead visualized telomeres using a strain carrying a *LacO* array integrated into the *sod2*⁺ locus in the subtelomeric region and also expressing LacI-GFP. The strain also contained RFP-tagged Cut11 as an NE marker (Ebrahimi, Masuda et al. 2018). Our new data shown in Supplementary Figs S7c and S7d) demonstrated that telomeres are significantly dissociated from the NE, similar to using Taz1-GFP.

Comment 4: I find it intriguing that Sad1 appears to be required for the association of telomeres with the nuclear envelope, as shown in Figure 4, whereas in Figure 5 the effect of Sad1 on telomere organization at the nuclear envelope appears to be modest. The authors should comment on these two results, to satisfy the reader. The authors should also prove the reproducibility of their *ura4+* reporter assays shown in Figures 5 A and B, to confirm the effect of the Sad1 5R mutant.

Thank you for the suggestion. First, our data indeed showed that telomeres disassociate from the nuclear envelope in the *sad1-5R* mutant (Figs. 4a and 4b). We repeated the experiments in Figure 5a (see Fig S9e). The results consistently showed that silencing of telomeres in the *sad1-5R* mutant is reduced. We also repeated the experiments in Figure 5b using by colony color (YES-ade medium) and growth rate (PMG-ade medium) (new Fig. 5b). The data also confirmed the silencing defects. Although the telomere (*tel2*) silencing loss is relatively modest in the *sad1-5R* mutant (Fig. 5a), the silencing loss of telomeres in the minichromosome 16 was obvious (new Fig. 5b).

Second, we want to point out that there is no direct correlation between telomere-NE association and telomere silencing. Previous studies show that the telomere localization does not directly affect telomere silencing (Chikashige, Haraguchi et al. 2010). Thus, the silencing defects observed in the *sad1-5R* mutant is not directly from the dissociation of telomeres from NE. According to our model, the Sad1-histone interaction plays an important role in the HDAC pathway to maintain the correct epigenetic state of heterochromatin regions, which may account for the silencing defect in the *sad1-5R* mutant. In the revision, we have provided new data showing that (1) *sad1-5R* has decreased Sir2 enrichment in telomeres (Fig. 5h); (2) the double mutants of *sad1-5R* with *clr3Δ* or *sir2Δ* exhibits no synthetic effect in heterochromatin silencing (Fig. S9d and S9e), supporting the idea that Sad1 acts in the same pathway as the HDACs. Thus, we believe the telomere silencing loss in *sad1-5R* stems from defective HDAC pathway, not from telomere dissociation from NE.

Comment 5: In Figure 6G, the authors did not conduct FRAP analysis on large puncta, such as those shown in the bottom panel of Figure 6F or Figure 6I. I would suggest to the authors that performing photobleaching on a larger area would make the quantification more precise and easier.

As the reviewer suggested, we also conducted FRAP analysis on the large area containing large puncta (Reference Fig. 5). The new data showed that the Sad1-GFP signal also has quick recovery after photobleaching, similar to the FRAP on the small puncta in Fig. 6g.

Reference Figure 5. The FRAP of overexpressed Sad1-GFP on a large puncta.

- a. Fluorescence images of overexpressed Sad1-GFP region used for FRAP experiment. The bleached region is indicated with a red square. Scale bars, 2 μm .
- b. The plot of normalized recovery of the overexpressed Sad1-GFP signal after photobleaching.

Comment 6: There appears to be a discrepancy between the quantitative plot in Figure 6J and the fluorescence recovery after photobleaching shown in Figure 6I. The authors should provide an explanation for this discrepancy or repeat the quantification.

As we explained above, in Fig 6i, Sad1-GFP was expressed at an endogenous level. The pre-beach Sad1-GFP signal on the NE is already weak compared to the overexpressed signal in Fig 6g (Note: Fig 6g and Fig 6i do not have the same contrast). Please note that taking a picture every 5 seconds during FRAP is a kind of photo-bleaching (although less intense), so this weak Sad1-GFP signal is subject to decay due to multiple photo-bleaching during the FRAP process. Therefore, the recovered signal after 40s are very weak compared to pre-bleach. The kind of imaging-induced intensity loss has significant effect on the weak fluorescence spots as seen in Fig. 6i, but the strong fluorescence spots are less affected as seen in Fig. 6d and 6g. On the other hand, the fluorescence recovery curve shown in our figures were normalized, thus appearing not to exactly match the image in Fig. 6i. Similar results have also been shown previously (Ebrahimi, Masuda et al. 2018).

Comment 7: Error bars should be added to Figures 6E, 6H, and 6J to represent statistical significance in figures.

As mentioned above, we repeated the FRAP experiments at least three times, and errors bars were added to new Fig 6e, 6h, and 6j.

References

- Bilokapic, S., M. Strauss and M. Halic (2018). "Histone octamer rearranges to adapt to DNA unwrapping." *Nat Struct Mol Biol* **25**(1): 101-108.
- Chikashige, Y., T. Haraguchi and Y. Hiraoka (2010). "Nuclear envelope attachment is not necessary for telomere function in fission yeast." *Nucleus* **1**(6): 481-486.
- Ebrahimi, H., H. Masuda, D. Jain and J. P. Cooper (2018). "Distinct 'safe zones' at the nuclear envelope ensure robust replication of heterochromatic chromosome regions." *Elife* **7**.
- Fernandez-Alvarez, A., C. Bez, E. T. O'Toole, M. Morpew and J. P. Cooper (2016). "Mitotic Nuclear Envelope Breakdown and Spindle Nucleation Are Controlled by Interphase Contacts

between Centromeres and the Nuclear Envelope." *Dev Cell* **39**(5): 544-559.

Ghanim, G. E., A. J. Fountain, A. M. van Roon, R. Rangan, R. Das, K. Collins and T. H. D. Nguyen (2021). "Structure of human telomerase holoenzyme with bound telomeric DNA." *Nature* **593**(7859): 449-453.

Hong, J., H. Feng, F. Wang, A. Ranjan, J. Chen, J. Jiang, R. Ghirlando, T. S. Xiao, C. Wu and Y. Bai (2014). "The catalytic subunit of the SWR1 remodeler is a histone chaperone for the H2A.Z-H2B dimer." *Mol Cell* **53**(3): 498-505.

Huang, Y., L. Sun, L. Pierrakeas, L. Dai, L. Pan, E. Luk and Z. Zhou (2020). "Role of a DEF/Y motif in histone H2A-H2B recognition and nucleosome editing." *Proc Natl Acad Sci U S A* **117**(7): 3543-3550.

Liang, X., S. Shan, L. Pan, J. Zhao, A. Ranjan, F. Wang, Z. Zhang, Y. Huang, H. Feng, D. Wei, L. Huang, X. Liu, Q. Zhong, J. Lou, G. Li, C. Wu and Z. Zhou (2016). "Structural basis of H2A.Z recognition by SRCAP chromatin-remodeling subunit YL1." *Nat Struct Mol Biol* **23**(4): 317-323.

London, N., B. Medina-Pritchard, C. Spanos, J. Rappsilber, A. A. Jeyaprakash and R. C. Allshire (2023). "Direct recruitment of Mis18 to interphase spindle pole bodies promotes CENP-A chromatin assembly." *Curr Biol* **33**(19): 4187-4201 e4186.

Sugiyama, T., H. P. Cam, R. Sugiyama, K. Noma, M. Zofall, R. Kobayashi and S. I. Grewal (2007). "SHREC, an effector complex for heterochromatic transcriptional silencing." *Cell* **128**(3): 491-504.

Wan, F. T., Y. B. Ding, Y. B. Zhang, Z. F. Wu, S. B. Li, L. Yang, X. Y. Yan, P. F. Lan, G. H. Li, J. Wu and M. Lei (2021). "Zipper head mechanism of telomere synthesis by human telomerase." *Cell Research* **31**(12): 1275-1290.

Wang, Y., S. Liu, L. Sun, N. Xu, S. Shan, F. Wu, X. Liang, Y. Huang, E. Luk, C. Wu and Z. Zhou (2019). "Structural insights into histone chaperone Chz1-mediated H2A.Z recognition and histone replacement." *PLoS Biol* **17**(5): e3000277.

Warren, C., J. B. Bonanno, S. C. Almo and D. Shechter (2020). "Structure of a single-chain H2A/H2B dimer." *Acta Crystallogr F Struct Biol Commun* **76**(Pt 5): 194-198.

REVIEWER COMMENTS

Reviewer #1 (Remarks to the Author):

The authors have addressed most of the criticisms and performed several suggested experiments to address critical points.

Reviewer #2 (Remarks to the Author):

Nature Communications #426048

While the reviewer appreciates the authors' effort to try to address the major points raised, there is still the lack of experimental evidence to support their major conclusions in a convincing way.

The Sad1-5R mutant has been used interchangeably as the lack of Sad1-H2A-H2B complex formation (or the lack of binding with free H2A-H2B for Sad1) in assessing cellular functions; however, this mutant has not been demonstrated to be defective in binding free H2A-H2B or nucleosomal H2A-H2B in vivo cell cultures. There might be more functions for this mutant other than the H2A-H2B binding as seen in vitro. Further, the newly added Fig. 5g is puzzling as according to the authors, the Sad1-5R mutant should NOT bind H2A-H2B, then how come Sad1-5R still forms complex with the H2AB as the ligand for Clr3 binding?

This also applies to all subsequent data. It is understandable that answering these questions require additional efforts (the authors used 'beyond the scope of current study' as the excuse to not pursue these questions), but minimal evidence is to show that the Sad1-5R only lacks free H2A-H2B binding function in cells without affecting other functions such as nuclear envelope assembly, nuclear shape, the LINK complex formation, etc. in cell cultures. Otherwise, the model and the conclusion (such as lines 33-35 in the Abstract, and lines 92-97) has to be revised to tune down its implication, which will unfortunately reduce the significance of the findings.

Other minor points:

For FRAP analysis in Fig. 6i, the authors argued that photo-bleaching during FRAP could have caused the lack of recovery of fluorescence, which does not seem to be true, as the other dot in the same image did not show much fluorescence reduction during the FRAP procedure (which should if indeed photobleaching was killing/reducing the fluorescence). Again, the fluorescence recovery of the representative image is minimal, which does not match the quantitation data in Fig. 6j.

The droplet sizes for H2AB-RFP in Fig. 6k and 6n are different, which should be the same to determine if indeed the Sad1-5R mutant is not enhanced by H2AB in terms of droplet formation.

Reviewer #3 (Remarks to the Author):

The authors have addressed all my points in detail and additional experiments were carried out. Overall, I am satisfied with the revisions and feel that the quality of the revised manuscript has been improved.

Minor Comment:

v Please indicate the abbreviation for inner nuclear membrane (INM) on P3 I59, i.e. the first time this term is quoted, and not on P4 I72.

Reviewer #1 (Remarks to the Author):

The authors have addressed most of the criticisms and performed several suggested experiments to address critical points.

We thank this reviewer for the appreciation of our work.

Reviewer #2 (Remarks to the Author):

While the reviewer appreciates the authors' effort to try to address the major points raised, there is still the lack of experimental evidence to support their major conclusions in a convincing way.

We thank the reviewer for the encouraging comment. The reviewer also still raised some concerns that we addressed below.

The Sad1-5R mutant has been used interchangeable as the lack of Sad1-H2A-H2B complex formation (or the lack of binding with free H2A-H2B for Sad1) in assessing cellular functions; however, this mutant has not been demonstrated to be defective in binding free H2A-H2B or nucleosomal H2A-H2B in vivo cell cultures. There might be more functions for this mutant other than the H2A-H2B binding as seen in vitro.

Thank you for the suggestion. To address the reviewer's concern, we created a strain carrying Sad1-5R-HA and H2B^{Htb1}-FLAG. Using this strain and also cells expressing H2B^{Htb1}-FLAG as a control, we conducted co-immunoprecipitation (co-IP) experiments. Our co-IP results demonstrated that indeed Sad1-5R is defective in binding histones in vivo (Reference Figure 1 and also shown in Supplementary Figure 4c).

Reference Figure 1. **Co-immunoprecipitation assays showed that the Sad1-5R mutant is unable to interact with histone H2B^{Htb1}.** Lysates from cells carrying Sad1-5R-HA and H2B^{Htb1}-FLAG were immunoprecipitated with an antibody specific for HA. Immunoprecipitated samples were analyzed by immunoblotting using anti-FLAG and anti-HA antibodies. Cells expressing H2B^{Htb1}-FLAG were used as a control.

Further, the newly added Fig. 5g is puzzling as according to the authors, the Sad1-5R mutant should NOT bind H2A-H2B, then how come Sad1-5R still forms complex with the H2AB as the ligand for Clr3 binding?

We are sorry for being unclear and would like to explain the “puzzling” data in Figure 5g as the following: Figure 5g measures the binding affinity between Clr3 and Sad1 by MST assay. The

binding reaction is measured by a fixed 10 nM fluorescently-labeled Clr3 (protein) and varied Sad1 concentration (ligand) in the MST capillaries. First, we measured the binding affinity between Clr3 and Sad1 alone (red curve in Fig. 5g) and obtained the K_d around 41.7 μM . To test the H2AB effect on Clr3-Sad1 binding, we mixed an equal amount of H2AB with Sad1 and then measured the binding between fluorescently-labeled Clr3 and the Sad1/H2AB mixture. We found that the presence of H2AB enhanced Sad1-Clr3 binding around 4-fold ($K_d=10.7 \mu\text{M}$, blue curve in Fig. 5g). For the Sad1-5R mutant, we also mixed the equal amount of H2AB into the Sad1-5R solutions (albeit they did not form a complex) and then measured the binding between Clr3 and the Sad1-5R/H2AB mixture. As the reviewer noted, the binding affinity between Clr3 and the Sad1-5R/H2AB mixture is same as the one between Clr3 and the Sad1 without histone H2AB, but lower than the one between Clr3 and the Sad1/H2AB mixture (Fig. 5g), thus indicating no enhanced binding for the Sad1-5R mutant to H2AB (Figure 5g). To avoid confusion, we have changed the labels in Fig. 5g to clearly state the ligand nature, and revised the manuscript to better describe the phenomenon: “but the addition of H2AB failed to increase the interaction between Sad1-5R and Clr3” (See Page 13, line 289).

This also applies to all subsequent data. It is understandable that answering these questions require additional efforts (the authors used ‘beyond the scope of current study’ as the excuse to not pursue these questions), but minimal evidence is to show that the Sad1-5R only lacks free H2A-H2B binding function in cells without affecting other functions such as nuclear envelope assembly, nuclear shape, the LINK complex formation, etc. in cell cultures. Otherwise, the model and the conclusion (such as lines 33-35 in the Abstract, and lines 92-97) has to be revised to tune down its implication, which will unfortunately reduce the significance of the findings.

Thank you for suggesting this. To determine whether the *sad1-5R* mutant affects the nuclear envelope and the nuclear shape, we examined the mutant strain carrying the GFP-tagged nucleoporin Nup189. We found that the distribution of Nup189-GFP in wild type and the *sad1-5R* mutant has no significant difference (Reference Figure 2), indicating that the mutant has little effect on the nuclear envelope. The figure also showed that the nuclear shape in the *sad1-5R* mutant has no significant difference compared to the wild type (Reference Figure 2).

Reference Fig. 2. Nup189-GFP distribution in the *sad1-5R* mutant. Wild-type cells carrying Nup189-GFP were used as a control. More than 200 cells for each were analyzed. Scale bar, 2 μ m.

The LINK complex in fission yeast, including the SUN domain Sad1 and the KASH domain protein Kms2, is essential for viability^{1,2}. However, our *sad1-5R* mutant is viable, suggesting that the LINK complex formation is not affected. In addition, Sad1 interacts with KASH domain proteins through its C-terminus domain in the intermembrane space, while the N-terminus domain of Sad1 is exposed to the nucleoplasm^{3,4}. The *sad1-5R* mutations we created are localized at the N-terminus domain of Sad1 in the nucleoplasm, and unlikely to have significant impact on its interaction with KASH domain proteins. Furthermore, the LINK complex in fission yeast plays an important role in centromere clustering and the spindle pole body (SPB) structure. Our data showed that the *sad1-5R* mutant has little effect on centromere clustering (Supplementary Figure 7a). To determine whether the SPB is affected in the mutant, we further analyzed the *sad1-5R* mutant carrying the mCherry-tagged SPB protein Sid4. We found that there is no significant difference in Sid4-mCherry distribution between wild type and the mutant (Reference Figure 3). Together, our data indicate that the Sad1-5R mutant that lacks free H2A-H2B binding acts in cells without affecting functions such as nuclear envelope assembly, nuclear shape, and the LINK complex formation.

Reference Fig. 3. Sid4-mCherry distribution in the *sad1-5R* mutant. Wild-type cells carrying Sid4-mCherry were used as a control. More than 200 cells for each were analyzed. Scale bar, 2 μ m.

Other minor points:

“For FRAP analysis in Fig. 6i, the authors argued that photo-bleaching during FRAP could have caused the lack of recovery of fluorescence, which does not seem to be true, as the other dot in the same image did not show much fluorescence reduction during the FRAP procedure (which should if indeed photobleaching was killing/reducing the fluorescence). Again, the fluorescence recovery of the representative image is minimal, which does not match the quantitation data in Fig. 6j.”

The point is well taken. Because Fig. 6i presents the Sad1 signal at the endogenous level, the fluorescence is much weaker than the overexpressed Sad1-GFP signals in Fig. 6g, and is more susceptible to photobleaching; Sad1 puncta also undergo dynamic movement, and there was thus a large deviation in the quantitation data as shown in Fig. 6j. The discrepancy in the previous Fig. 6i and 6j is likely due to the fact that the single fluorescence image represented the low point of the quantitation data compared to the averaged FRAP curve. To avoid confusion, we now provided a new FRAP figure in Fig. 6i in the revised manuscript, of which the fluorescence recovery nearly matches the averaged curve.

“The droplet sizes for H2AB-RFP in Fig. 6k and 6n are different, which should be the same to determine if indeed the Sad1-5R mutant is not enhanced by H2AB in terms of droplet formation.”

The droplet sizes for H2AB-RFP are indeed different in Fig. 6k and 6n. Please note that Fig. 6k and 6n are the droplet images of Sad1-GFP mixed with H2AB-RFP, not Sad1 or H2AB alone. When H2AB-RFP is mixed with WT Sad1 (Fig. 6k), H2AB and Sad1 are in the same droplet (green and red are fused together) and the droplet size is much larger than Sad1 alone or H2AB alone (Fig. 6l). When H2AB-RFP is mixed with Sad1-5R-GFP (Fig. 6n), due to the no interaction between H2AB and Sad1-5R, the green droplet (Sad1-5R) and the red droplets (H2AB) separated. More importantly, the size of H2AB-RFP droplets in the Sad1-5R/H2AB mixture remained the same as H2AB-RFP alone (Fig. 6l and Supplementary Fig. 11a) which was much smaller than the H2AB-RFP size in the Sad1-WT/H2AB mixture.

Reviewer #3 (Remarks to the Author):

The authors have addressed all my points in detail and additional experiments were carried out. Overall, I am satisfied with the revisions and feel that the quality of the revised manuscript has been improved.

We thank this reviewer for the appreciation of our work.

Minor Comment:

Please indicate the abbreviation for inner nuclear membrane (INM) on P3 159, i.e. the first time this term is quoted, and not on P4 172.

Thanks. This suggestion has been taken.

References:

- 1 Hagan, I. & Yanagida, M. The product of the spindle formation gene *sad1+* associates with the fission yeast spindle pole body and is essential for viability. *J Cell Biol* **129**, 1033-1047, doi:10.1083/jcb.129.4.1033 (1995).
- 2 Kim, D. U. *et al.* Analysis of a genome-wide set of gene deletions in the fission yeast *Schizosaccharomyces pombe*. *Nat Biotechnol* **28**, 617-623, doi:10.1038/nbt.1628 (2010).
- 3 Tapley, E. C. & Starr, D. A. Connecting the nucleus to the cytoskeleton by SUN-KASH bridges across the nuclear envelope. *Current opinion in cell biology* **25**, 57-62, doi:10.1016/j.ceb.2012.10.014 (2013).
- 4 Fernandez-Alvarez, A., Bez, C., O'Toole, E. T., Morphew, M. & Cooper, J. P. Mitotic Nuclear Envelope Breakdown and Spindle Nucleation Are Controlled by Interphase Contacts between Centromeres and the Nuclear Envelope. *Dev Cell* **39**, 544-559, doi:10.1016/j.devcel.2016.10.021 (2016).

REVIEWERS' COMMENTS

Reviewer #2 (Remarks to the Author):

The authors addressed my concerns.

REVIEWERS' COMMENTS

Reviewer #2 (Remarks to the Author):

The authors addressed my concerns.

We thank this reviewer for the appreciation of our work and constructive suggestions.